# Multivariate analysis of electrophysiological diversity of *Xenopus* visual neurons during development and plasticity

Christopher M Ciarleglio[1†], Arseny S Khakhalin[1,2†], Angelia F Wang[1], Alexander C Constantino[1], Sarah P Yip[1], Carlos D Aizenman[1]*

[1]Department of Neuroscience, Brown University, Providence, United States; [2]Biology Program, Bard College, Annandale-on-Hudson, United States

**Abstract** Biophysical properties of neurons become increasingly diverse over development, but mechanisms underlying and constraining this diversity are not fully understood. Here we investigate electrophysiological characteristics of *Xenopus* tadpole midbrain neurons across development and during homeostatic plasticity induced by patterned visual stimulation. We show that in development tectal neuron properties not only change on average, but also become increasingly diverse. After sensory stimulation, both electrophysiological diversity and functional differentiation of cells are reduced. At the same time, the amount of cross-correlations between cell properties increase after patterned stimulation as a result of homeostatic plasticity. We show that tectal neurons with similar spiking profiles often have strikingly different electrophysiological properties, and demonstrate that changes in intrinsic excitability during development and in response to sensory stimulation are mediated by different underlying mechanisms. Overall, this analysis and the accompanying dataset provide a unique framework for further studies of network maturation in Xenopus tadpoles.

*For correspondence: Carlos_Aizenman@brown.edu

[†]These authors contributed equally to this work

**Competing interests:** The authors declare that no competing interests exist.

## Introduction

Electrophysiological properties of neurons become increasingly diverse over development in ways that are critical for proper nervous system function and maturation (*Turrigiano and Nelson, 2004*; *Marder and Goaillard, 2006*). Perturbation of these processes can have broad and devastating consequences leading to neurodevelopmental disorders such as mental retardation, autism, and schizophrenia (*Rice and Barone, 2000*; *Belmonte et al., 2004*; *Pratt and Khakhalin, 2013*). It remains unclear, however, to what degree this diversity in electrophysiological tuning reflects intrinsic developmental differentiation, and how much it reflects the particular activation history of a given neuron, as well as the constraints that shape how well neurons adapt to changes in their input patterns.

The adaptability of electrophysiological properties is central for allowing developing neural circuits to maintain functional stability, while simultaneously providing flexibility for accommodating developmental changes. One mechanism that contributes to this balance is homeostatic plasticity, whereby neurons adjust their synaptic and intrinsic properties based on the activity of the circuit in which they are embedded (*Daoudal and Debanne, 2003*; *Desai, 2003*; *Turrigiano and Nelson, 2004*; *Ibata et al., 2008*; *Turrigiano, 2008*; *Marder, 2011*). Homeostatic plasticity allows developing circuits to function stably by maximizing their dynamic range as new inputs become incorporated (*Bucher et al., 2005*; *Marder and Goaillard, 2006*; *Pratt and Aizenman, 2007*). This is particularly

**eLife digest** Brains consist of many cells called neurons: billions of them in a human brain, and hundreds of thousands in the brain of a small fish or a frog tadpole. Many of these neurons are very much alike, and work together to process information in the brain. Yet while they are similar, they are not exactly identical. One of the reasons for these differences seems to be to allow each neuron to contribute something unique to the overall working of the brain. By looking at how individual neurons within a specific type differ from each other, it is possible to understand more about how they work together.

Ciarleglio, Khakhalin et al. have now compared the properties of the neurons in a part of the brain of a developing frog tadpole that processes sensory information. This showed that these neurons appear relatively similar to each other in young tadpoles. However, as the tadpoles grow and their brains become more elaborate the neurons become increasingly diverse, and their properties become more unique and nuanced.

One possible explanation is that this diversity reflects new types of neurons being formed; another, that the differences between the neurons reflect how these cells have adapted to different patterns of sensory input they may have experienced. To distinguish between these two possibilities, Ciarleglio, Khakhalin et al. provided a group of older tadpoles with strobe-like visual stimulation and observed that this caused the neurons to become more similar once again. This suggests that neurons can change their response properties to adapt to the type of sensory input they receive, which would allow the animal to better process different types of sensory information. The data collected through these experiments could now be used to build computational models of this part of the tadpole brain.

relevant to developing animals: their nervous system must be functional and able to interact with its environment even as nascent circuitry is still developing.

One place where this adaptability in synaptic and intrinsic properties is particularly salient, is in the optic tectum of *Xenopus laevis* tadpoles—a midbrain area that processes inputs from visual, auditory, and mechanosensory systems (*Cline, 1991*; *Ewert, 1997*; *Cline, 2001*; *Ruthazer and Cline, 2004*; *Ruthazer and Aizenman, 2010*). Sensory inputs to the tectum are strengthened over development, resulting in increasingly robust synaptic responses, yet this strengthening is accompanied with decreases in intrinsic excitability that may function to maintain a stable dynamic range in this circuit (*Pratt and Aizenman, 2007*). As a consequence, visually guided behaviors, such as collision avoidance, improve and become more tuned to specific stimuli (*Dong et al., 2009*). Changes in sensory environment can also elicit homeostatic plasticity in tectal cells, resulting in adjustment of both synaptic and intrinsic properties (*Aizenman et al., 2003*; *Deeg and Aizenman, 2011*).

Since homeostatic plasticity coordinates changes of different cellular properties, over time it is expected to constrain these properties, limiting ways in which they can co-vary within the population of cells (*O'Leary et al., 2013*): for example, strong excitatory synaptic drive results in lower intrinsic excitability. Coordinated changes in different physiological properties may contribute to diversification of cell tuning that happens as networks mature, creating and shaping differences in cell phenotypes both between cell types as they emerge (*Ewert, 1974*; *Frost and Sun, 2004*; *Kang and Li, 2010*; *Nakagawa and Hongjian, 2010*; *Liu et al., 2011*), and within each cell type in a functional network (*Tripathy et al., 2013*; *Elstrott et al., 2014*). These considerations suggest that multivariate distributions of different physiological properties sampled across many cells in a network may contain unique information both about current tuning of this network, and the mechanisms behind this tuning that may act through local recalibration of properties in individual cells (*O'Leary et al., 2013*). Yet relatively few studies have attempted this kind of analysis on a large scale so far.

Here we perform a large-scale electrophysiological census of retinorecipient neurons in the developing *Xenopus laevis* tectum to better understand the electrophysiological variability of tectal neurons in development, and in response to a need for homeostatic change. Using a comprehensive suite of tests we describe relationships between 33 electrophysiological variables, and show that both the variability and the predictability of multivariate cell tuning increases over development, and

undergo changes in response to sensory stimulation. By analyzing groups of neurons that produce similar spike trains, we also show that similar spiking behaviors may be supported by different combinations of underlying electrophysiological properties.

## Results

### Main dataset and correlation analysis

We recorded from 155 deep-layer, retinorecipient tectal cells across developmental stages 43 to 49 (*Nieuwkoop and Faber, 1994*) from 42 animals, measuring from 9 to 33 different electrophysiological variables in each cell (median of 26 variables per cell; see *Figure 1* and *Supplementary file 1* for a graphical description and a concise table of variables, and the Materials and methods for a detailed description of each variable). Of 155 cells, 35 cells contributed to all 33 variables, 62 contributed to 30 variables, 124 to 20 variables, and 154 to 10 variables. Across different variables, the least covered variable had measurements from 64 cells, while the most covered one had measurements from 154 cells (median of 134 cells per variable); in total, 18% of all possible observations were missing. The dataset containing analysis parameters is available online as *Supplementary file 2*. The entire dataset including raw electrophysiology files has also been made available and can be accessed with the following doi:10.5061/dryad.18kk6.

With 33 different cell parameters, we had 528 potential pairwise correlations to assess. Ninety of these correlations were significant after FDR adjustment ($\alpha = 0.05$, corresponding to Pearson correlation p-value threshold of about 8e−3; *Figure 2A*); all correlations with $r > 0.5$ (n=11) were also significant for Spearman correlation after same FDR adjustment. We found that there were at least three distinct clusters of tightly correlated variables representing different neural characteristics: tectal cell spikiness (a tendency to produce more spikes); measurements of spike shape and dynamics that strongly anti-correlated with spikiness; and intrinsic neuronal properties, such as membrane capacitance and amplitudes of intrinsic ionic currents. Some of the more salient correlations are shown in *Figure 2B*.

The presence of correlations in our dataset suggested that not all variables were independent, which is expected for sets of variables that describe different aspects of common underlying cellular qualities, such as spikiness or synaptic effectiveness (*McGarry et al., 2010*). To make sure that none of the variables were redundant (brought no new information to the set), or too noisy (having no interactions with other variables in the set), we ran the so-called Principal Variable Analysis, quantifying the total amount of variance in the dataset explained by each variable (*Mccabe, 1984*). We found that the most informative, and thus least independent, variables were related to the number of spikes the cell was able to generate (N spikes to cosine and step injections explained 15% and 14% of total variance, respectively), or different aspects of spike shape (9–13% of total variance; *Figure 2C*, *top* of the list). Conversely, some variables did not serve as good linear predictors for other properties in the set (*bottom* of *Figure 2C*), suggesting that they were more independent — the lowest variable explained 4% of total variance, which was still more than the 3% expected for a fully uncorrelated variable. In summary, it can be seen from *Figure 2C* that no single variable was 'too good' in explaining overall variance in the dataset, but also that none of them fell at or below the predicted noise level; there was no clear division of variables into distinct groups, but rather a smooth decline across the list. Finally, variables of different biological nature were diversely distributed across the parts of the plot corresponding to higher and lower explanatory power. From this we concluded that our dataset was well balanced for exploratory analysis, offering a healthy mix of independent and interacting variables (*Guyon and Elisseeff, 2003*).

Knowing that the maximal number of spikes in response to current step injections can alone explain 14% of total variance in the dataset made us wonder which protocol of those we used was the most informative in the sense of best capturing the electrophysiological identity of each tectal cell. To answer this question, we ran the Principal Variance Analysis on a combination of variables coming from different protocols. We found that 9 variables from the step current injection protocol together explained 42% of total variance in the set; 7 variables quantifying IV-curves explained 34%; 6 variables from cosine current injections explained 33%; 5 synaptic variables and 4 passive membrane properties both accounted for 21%; and 2 variables describing miniature postsynaptic currents (mEPSCs) predicted 11% of total variance (shares of explained variance don't have to add

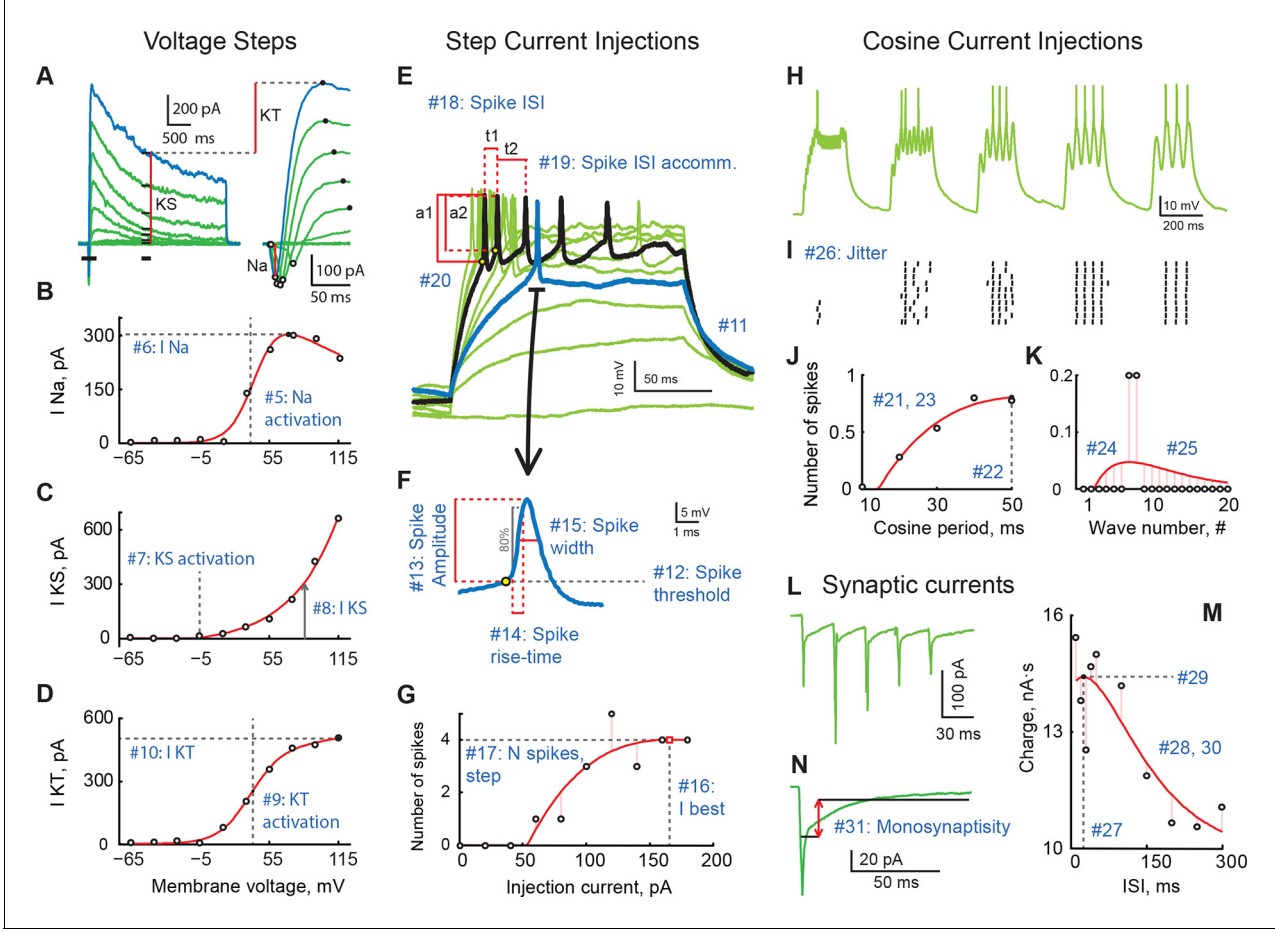

**Figure 1.** A review of cell properties characterized in this study (See methods for a detailed description of every measurement). (A) Response to voltage clamp steps after passive current subtraction; full currents on the *left*, a zoom-in look at early active currents on the *right*. Red vertical lines show how voltage-gated sodium ($I_{Na}$), stable potassium ($I_{KS}$), and transient potassium currents ($I_{KT}$) were measured. (B) IV curve for $I_{Na}$. (C) IV curve for $I_{KS}$. (D) IV curve for $I_{KT}$. (E) Spiking responses to current step injections of different amplitudes; first response to produce a spike is shown in **blue**; response generating maximal number of spikes is shown in **black**. (F) An expanded look at the first spike produced by the cell in response to step current injections. (G) Number of spikes as a function of step current injection amplitude. (H) A trace of membrane potential recorded from the cell in response to cosine current injections of varying frequency. (I) Spike-raster of 10 consecutive responses to cosine injections shown in H. (J) Average number of spikes in response to a single cosine injection as a function of cosine period. (K) Average number of spikes per cosine wave in response to injections of shortest period (10 ms; *leftmost* group in panels H and I). (L) Sample trace of excitatory synaptic currents recorded in voltage clamp mode in response to optic chiasm stimulation with a 30 ms inter-stimulus interval. (M) Total postsynaptic charge as a function of inter-stimulus interval. (N) Average postsynaptic current showing time-windows that were used to build the "Monosynapticity ratio". Panels (A–D) originate from one cell; panels (E–M) were recorded in another cell; both from stage 49 animals.

up to 100%). Based on this analysis, we conclude that the step current injection protocol (see *Figure 1E–G*) is thus one of the most informative, and should be recommended for fast profiling of cell types in the future.

## Changes with development

We next asked which electrophysiological characteristics changed with development as tadpoles matured from stage 43, a time when axons from the eye make first functional connections in the brain (*Holt and Harris, 1983*), to stage 49, when tectal networks become sufficiently refined to support spatially coordinated visual behaviors and multisensory integration (*Deeg et al., 2009*; *Dong et al., 2009*; *Xu et al., 2011*; *Khakhalin et al., 2014*). To compensate for the unevenness of data availability across cell properties and developmental stages, we combined data from stages 43–44, 45–46, and 48–49, as these groupings have proven useful in previous studies (*Pratt et al.,*

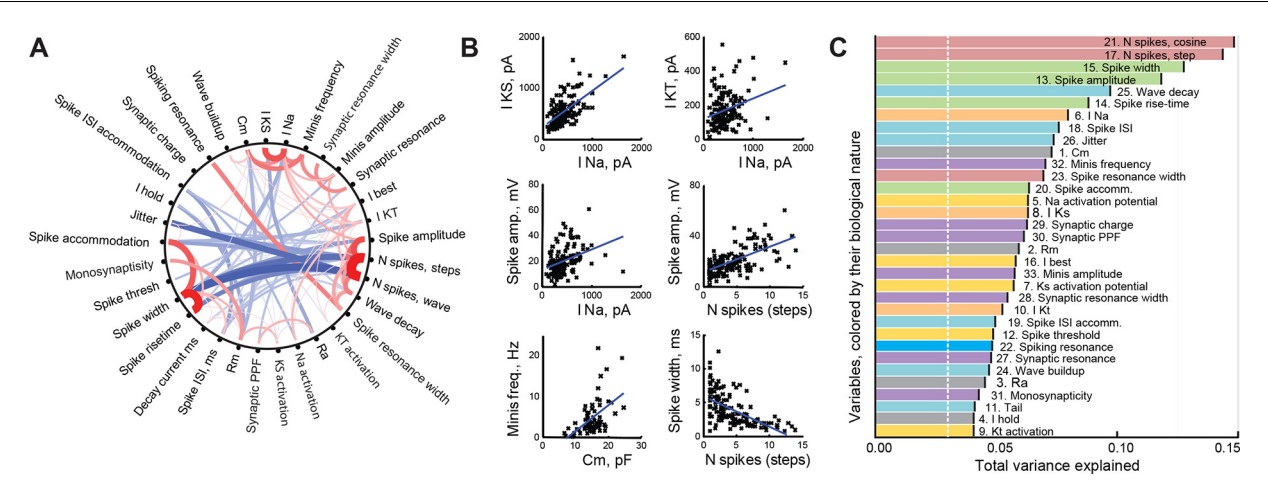

**Figure 2.** Correlations between cell properties. (A) Diagram illustrating significant correlations and their strengths (r-values) for all pairs of cell properties measured. Positive correlations are shown in red; negative correlations in **blue**; the width and darkness of each line are proportional to respective r-value. (B) Selected significant correlations between cell variables. (C) Variables sorted by the total variance each of them can explain in the full dataset (amounts are not additive and do not add up to 100%). Colors indicate the biological nature of each variable (red for spiking, **green** for spike shape, **blue** for spike timing, orange for ionic currents, yellow for activation potentials, gray for passive properties, and **purple** for synaptic properties). The white dashed line shows the expected variance explained for an ideally uncorrelated variable.

*2008*; *Dong et al., 2009*; *Sharma and Cline, 2010*). We did not group stage 47 with others however, as there were indications that this transient stage in development may be unique in terms of tadpole behavior, network excitability, and average tectal cell properties (*Pratt and Aizenman, 2007*; *Bell et al., 2011*). We found that 12 cell properties changed significantly across these developmental periods ($P_{ANOVA} < 0.05$), with 5 values decreasing overall, and 6 variables increasing (*Figure 3*; also see *Supplementary file 3* for summary table).

While average values changed in both directions during development, the effect on variability of cellular properties was much more consistent: 14 variables out of 33 increased their variability from stages 45–46 to 48–49 ($P_V < 0.05$, corresponding to increases in standard deviation of 40% and higher; see *Supplementary file 3*). Among others, spiking inactivation (as measured by 'Wave decay'), maximal amplitudes of sodium and slow potassium ionic currents, the frequency of minis, and the total synaptic charge all experienced an almost twofold increase in variability. Only two properties out of 33 became less variable over development: synaptic resonance and synaptic resonance width. These data suggest that by stages 48–49, neurons became more electrophysiologically diverse than they were at stages 45–46.

## Factor analysis

To visualize and explore the patterns behind co-dependence and co-variance of variables in our dataset, and to better measure common features that underlie these correlations, we used principal component analysis (PCA). As not every variable was measured in every cell, we used an iterative Bayesian version of PCA known as 'PCA with missing values' (*Ilin and Raiko, 2010*), followed by a promax oblique rotation. We extensively verified the validity of our PCA analysis, comparing it to standard PCA on restricted and imputed data, PCA on rank-transformed data, as well as two most common non-linear dimensionality reduction approaches: Isomap and Local Linear Embedding (see Materials and methods). We concluded that our PCA analysis was the most appropriate analysis for for this data set, and performed better than local non-linear approaches, with the first two principal components explaining 15% and 8% of total variance respectively (this total of 23% of variance explained would have corresponded to ~35% of variance if we had every type of observation in every cell; see Materials and methods for details).

A loading plot (*Figure 4A*) shows contributions of individual variables from the dataset to rotated PCA components. Points on the plot are colored according to the biological nature of each variable

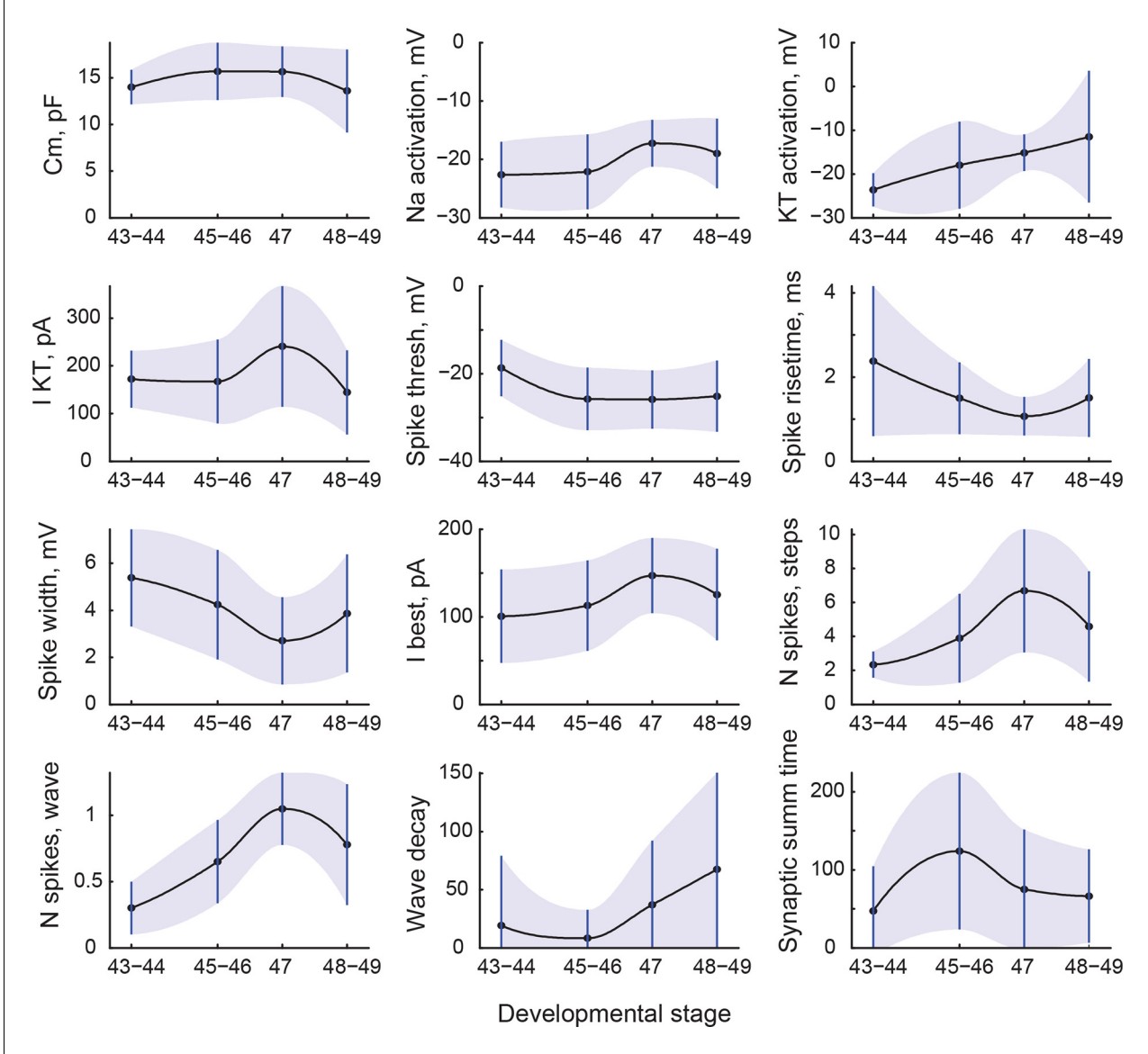

**Figure 3.** Changes in cell properties with age. All cell properties that significantly changed with development are shown here as mean values (central line) and standard deviations (whiskers and shading). Transitions between points are shown as shape-preserving piecewise cubic interpolations.

(*Figure 4*, see legend); variables shown on the right contributed positively to the first component (C1), while those on the left contributed negatively to this component; variables in the upper part of the cloud contributed positively to the second component (C2), while those at the bottom contributed negatively to it. Consistent with high predictive value of individual variables related to spiking, we found that C1 describes the overall 'Spikiness' of each cell. Cells with large values of C1 generated many sharp and narrow spikes in response to current injections (*Figure 4A*, green points describing spike shape, *left*; **red** points describing number of spikes, *right*). These cells spiked strongly in response to prolonged current injections (*Figure 4A*, red point for 'Spiking resonance width,' *upper right quadrant*); had low accommodation, both in spike amplitude (*Figure 4A*, green, *left bottom corner*) and inter-spike interval (*Figure 4A*, blue, *left top corner*), and had high trial-to-trial spike-timing precision (low jitter; *Figure 4A*, blue point, *left top corner*). Properties of cells with different C1 and C2 values can also be illustrated in a modified score-plot, in which traces of membrane potential responses to a 100 pA step current injection are arranged within the C1-C2 principal component space (*Figure 4B*). Note the difference in spiking responses between cells on *right* (high

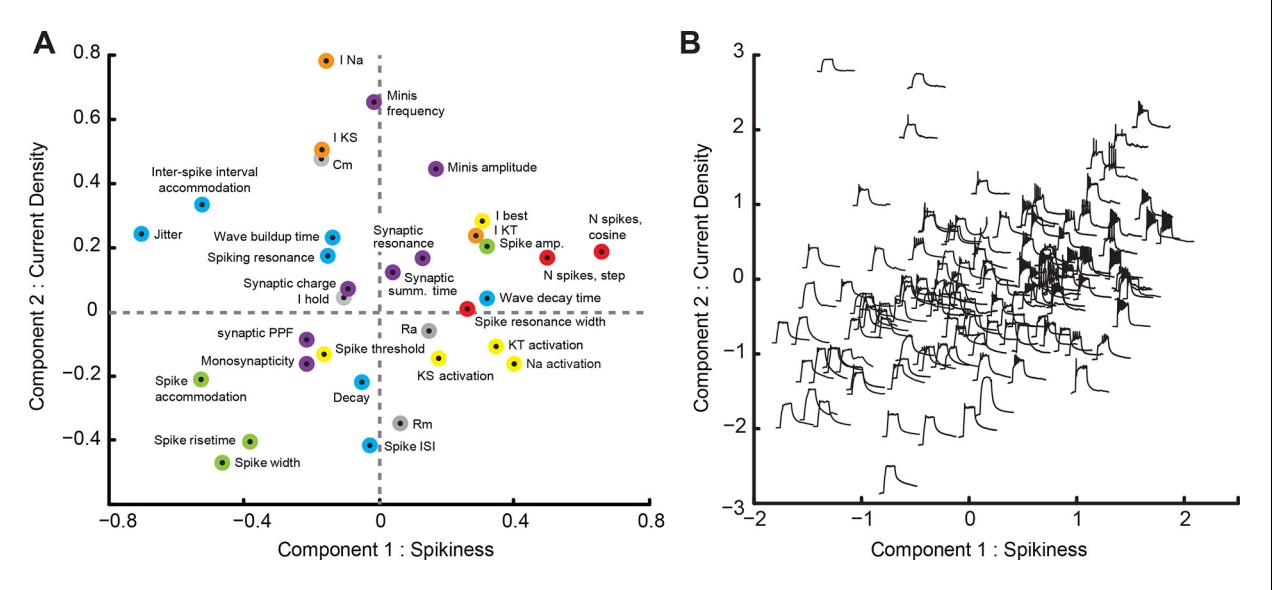

**Figure 4.** Principal Component Analysis (PCA). (**A**) Loading-plot, presenting contribution of individual cell properties to the first two PCA components (see detailed description in the text). Points are colored with regards to how they describe the spikiness of the cell (red), shape of spikes (green), their temporal properties (blue), ionic currents (orange), passive electrical properties (gray), or synaptic properties of the cell (purple). (**B**) Modified score-plot showing how individual cells score on first two PCA components, with responses of respective cells to step current injections used instead of standard plot markers. Responses on the *right* are spikier that those on the *left*, while responses on the *bottom* have a greater passive component than responses on the *top*.

C1) and *left* (low C1) sides of *Figure 4B*. Cells with large C1 also tended to be involved in polysynaptic networks (*Figure 4A*, purple point for 'Monosynapticity coefficient,' *lower left quadrant*) and did not exhibit short-term facilitation of synaptic inputs during repeated stimulation (*Figure 4A*, purple point for 'Synaptic PPF,' *lower left quadrant*). Cells with low values of C1 exhibited opposite traits: they produced few broad, squat, quickly accommodating spikes (*Figure 4B*, *left*), were not recruited in recurrent networks, and tended to have strong synaptic facilitation that could potentially indicate high plasticity of synaptic inputs (*Kleschevnikov et al., 1997*).

The second component (C2) can be loosely dubbed 'Current density': cells with high values of C2 had large intrinsic ionic currents (voltage-gated sodium $I_{Na}$ and slow potassium $I_{KS}$ currents), high membrane capacitance (Cm) and low membrane resistance (Rm), consistent with a larger membrane surface, and received strong synaptic inputs, in terms of both frequency and amplitude of mEPSCs. These cells produced frequent and sharp spikes (*Figure 4A*, blue point for 'Spike ISI' and **green** points for 'Spike rise-time' and 'Spike width,' *lower part of the plot*), but also tended to have higher values of spike-timing jitter and inter-spike interval accommodation. Conversely, cells with low values of C2 behaved as smaller cells electrophysiologically (low Cm, high Rm), and had weak intrinsic and spontaneous synaptic currents. As principal neurons in the optic tectum have relatively uniform geometrical cell body sizes (*Lazar, 1973*), these differences in electrophysiological properties may indicate different levels of electrical coupling between the cell bodies, where the recording was performed, and both dendritic and axonal compartments, where the currents are generated.

After classifying all cells according to their position in the PCA space (*Figure 5A*), we were able to visualize developmental maturation of tectal cells as movements of point clouds within the C1-C2 plane, and as changes of these clouds' shapes (*Figure 5B*). It can be seen from *Figure 5B* that as cells matured, their representations in the PCA space migrated from the *left* to the *right* side of the plot. Indeed, the *Spikiness* (C1) changed with age ($P_{ANOVA} = 1e{-}5$), first increasing from stages 43–44, through 45–46, and up to stage 47 ($P_{MW} < 4e{-}3$ at each transition, N = 11, 64 and 24 respectively), and then decreasing again at stages 48–49 ($P_{MW} = 0.02$, N = 24, 56; *Figure 5B*). The value of C2, or 'Current density', did not change much over development ($P_{ANOVA} = 0.08$) except for a slight decrease between stages 47 and 48–49 ($P_{MW} = 0.02$, N = 24, 56).

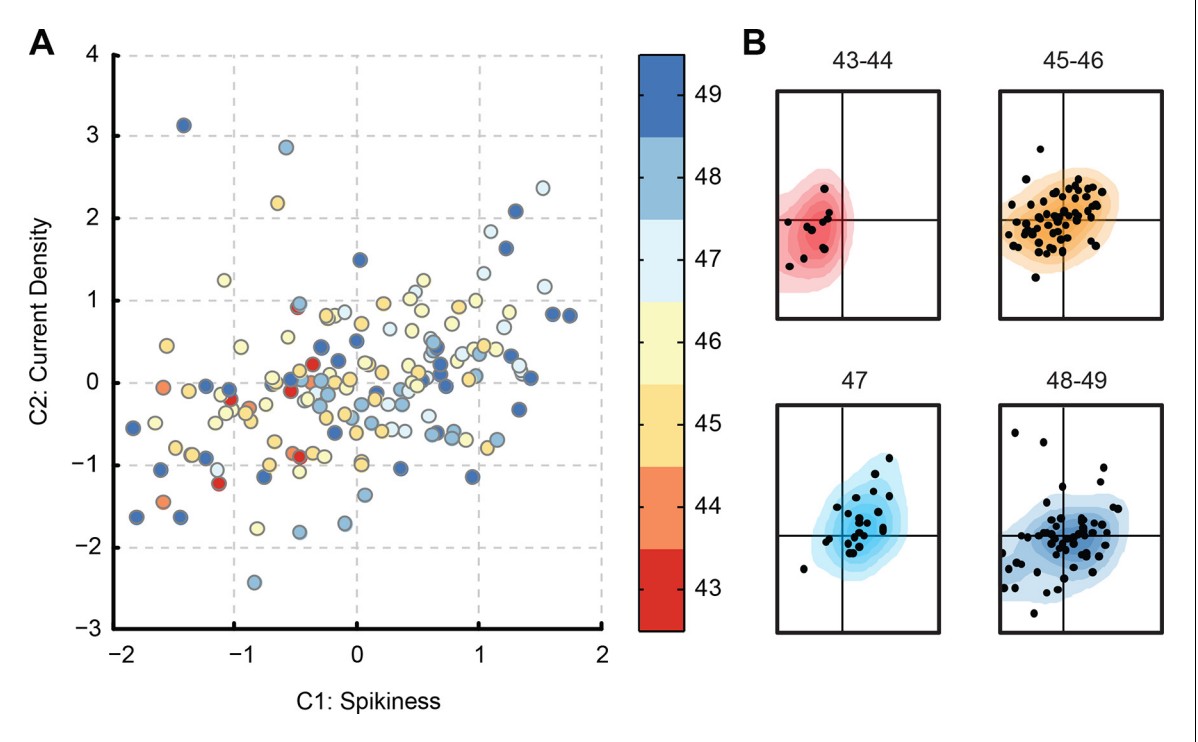

**Figure 5.** Evolution of PCA component scores with development. (A) Score-plot of PCA scores for all cells from the main dataset, with cells colored by the developmental stage of the animal: from *reds* for stages 43–44, through *yellows* for stages 45–46, to **blues** for stages 48–49. Note that most **red** cells are located on the *left*, while most light blue cells (stage 47) are located on the *right*. (B) Isolated sub-clouds of points from panel (A), shown on the same axes as panel (A), and illustrating the progression of different stages through the score-plot (N points = 11, 64, 24, and 56). Estimated density kernels for sub-clouds are shown as colored backgrounds. Stages 43–47 illustrate that the cloud moved to the *right*, while at stages 48–49 it moved back to the *center*, and expanded at all directions (see quantification in the text).

The clouds of points shown in *Figure 5B* also differ in size and structure, with the stages 45–46 cloud being more compact and simple, and the stages 48–49 cloud being more sprawled with an uneven distribution of points. To quantify cloud size, we compared medians of pairwise Euclidian distances between points in the PCA space and found that they were larger at stage 48–49 than at any other developmental group ($P_{MW} < 5e-7$ for all comparisons). Likewise, the value of 'Current density' (C2) was more variable at stages 48–49 than at stages 45–46 ($P_V = 0.004$, N = 64, 56), and the average pairwise difference between cells in the original 33-dimentional space was 22% larger at stages 48–49 than at stages 45–46 ($P_{TT} < 1e-11$; see Materials and methods for details). This suggests that from stages 42 to 47, neurons changed their properties consistently, gradually getting more spiky, but then scaling their average spikiness back at stages 48–49, while simultaneously expanding to the whole range of possible C1-C2 values, increasing the diversity of electrophysiological tuning across individual tectal cells.

We then quantified the presence of internal structure within the original 33-dimentional datasets, separately for younger and older cells, by performing multiple imputation on the data, and then running two different types of unsupervised feature analyses that identify structure in multivariate distributions: the hierarchical cluster analysis that looks for subclouds of points within a larger cloud, and local PCA (as opposed to previously described global PCA that was run on the full set of data) to quantify linear interdependencies between cell properties within each age group. For cluster analysis, we used the agglomerative nesting coefficient AGNES (*Struyf et al., 1996*) and found (*Figure 6A*) that the degree of clustering was 36% larger at stages 48–49 compared to stages 45–46 (0.67 ± 0.04 and 0.49 ± 0.03 respectively, $P_{TT} < 1e-11$), reflecting a substantial increase in within-group heterogeneity. For local PCA analyses, we looked at the amount of total variance explained by the first 2 components within each stage group as a measure of internal structure and linear

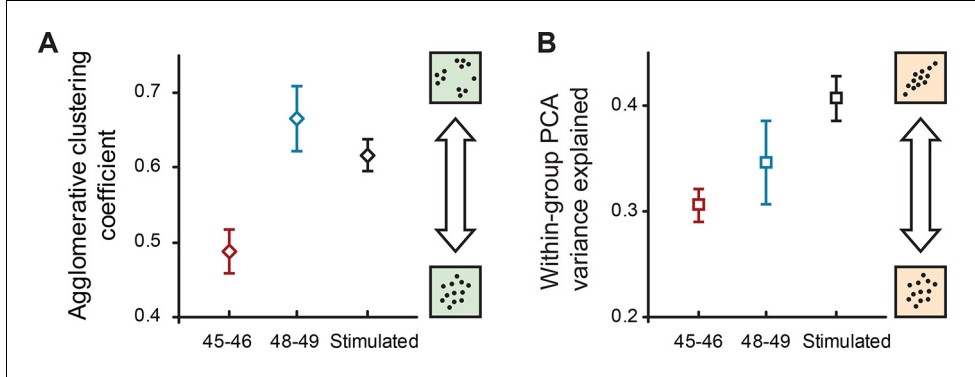

**Figure 6.** Internal structure of cell property distributions. (**A**) Agglomerative clustering coefficients for properties of naïve cells at stages 45–46, 48–49, and cells after sensory stimulation (stage 49). Higher values correspond to higher levels of clustering (grouping); lower values correspond to more Gaussian-like unimodal distributions. (**B**) The amount of within-group variance explained by the first two components of PCA for the same groups of data. Higher values correspond to higher correlations between different electrophysiological variables in the set. Both plots show means ± standard deviations of results obtained in the original 33-dimensional space after multiple imputation with subsampling.

interactions between different variables. We found (*Figure 6B*) that despite an increase in total variance and heterogeneity in older cells, locally run PCA explained a slightly higher degree of variance in these cells (0.35 ± 0.04) than in younger cells at stages 45–46 (0.31 ± 0.02; $P_{TT}$ < 1e−11) suggesting that in more mature networks, cell properties are more coordinated with each other.

## Effects of sensory stimulation

While our analysis of neuronal maturation demonstrated that several cellular properties changed over development, both in terms of their average values and their variability, we were not able to tell whether these phenomena represented genetically determined developmental programs, or if they reflected experience-dependent adaptations of electrophysiological properties resulting from the cumulative sensory experience of each cell (*Dong et al., 2009*; *Munz et al., 2014*). It is known from previous studies that tadpoles exposed for several hours to strongly patterned visual stimulation show synaptic (*Aizenman et al., 2002*; *Aizenman and Cline, 2007*), intrinsic (*Aizenman et al., 2003*; *Dong et al., 2009*) and morphological (*Sin et al., 2002*) changes in individual tectal cells, as well as in tectal network activity (*Pratt et al., 2008*), and animal behavior (*Dong et al., 2009*). We provided four hours of strongly patterned visual stimulation to an experimental group of stage 49 tadpoles and then measured and analyzed same 33 variables from 65 cells (across 19 animals).

Of 33 electrophysiological properties, 8 changed significantly between naïve and visually stimulated cells (see *Supplementary file 3*). As previously described (*Aizenman et al., 2003*), visually stimulated cells spiked more in response to step current injections, producing on average 7.1 ± 4.7 spikes per injection, compared to only 4.6 ± 3.3 spikes in naïve cells ($P_{MW}$ = 5e−3, N = 51, 60; *Figure 7A*). Interestingly, while for naïve cells the number of spikes produced in response to cosine injections strongly correlated with the number of spikes in response to step injections (r = 0.82, $P_{corr}$ = 2e−27, N = 108), visually stimulated cells did not change their spiking response to cosine injections (0.8 ± 0.5 for naïve, 0.9 ± 0.5 for stimulated cells; p = 0.3, N = 38, 60), indicating that changes in spikiness in development, and after visual experience, may be implemented by different biophysical mechanisms. Other variables that increased after stimulation included the absolute value of the holding current, and spike threshold potential (*Figure 7B*). Three properties decreased after visual stimulation: membrane capacitance (*Figure 7C*), speed of response build-up during cosine stimulation ("Wave build-up"), and jitter.

*Figure 7D* shows $\eta^2$ effect sizes and marks statistical significance for changes during development and in response to sensory stimulation (*Ferguson, 2009*). It can be seen that the overlap between normal changes in development and changes after visual stimulation is very small: only 3 electrophysiological properties changed significantly both in development and after stimulation —

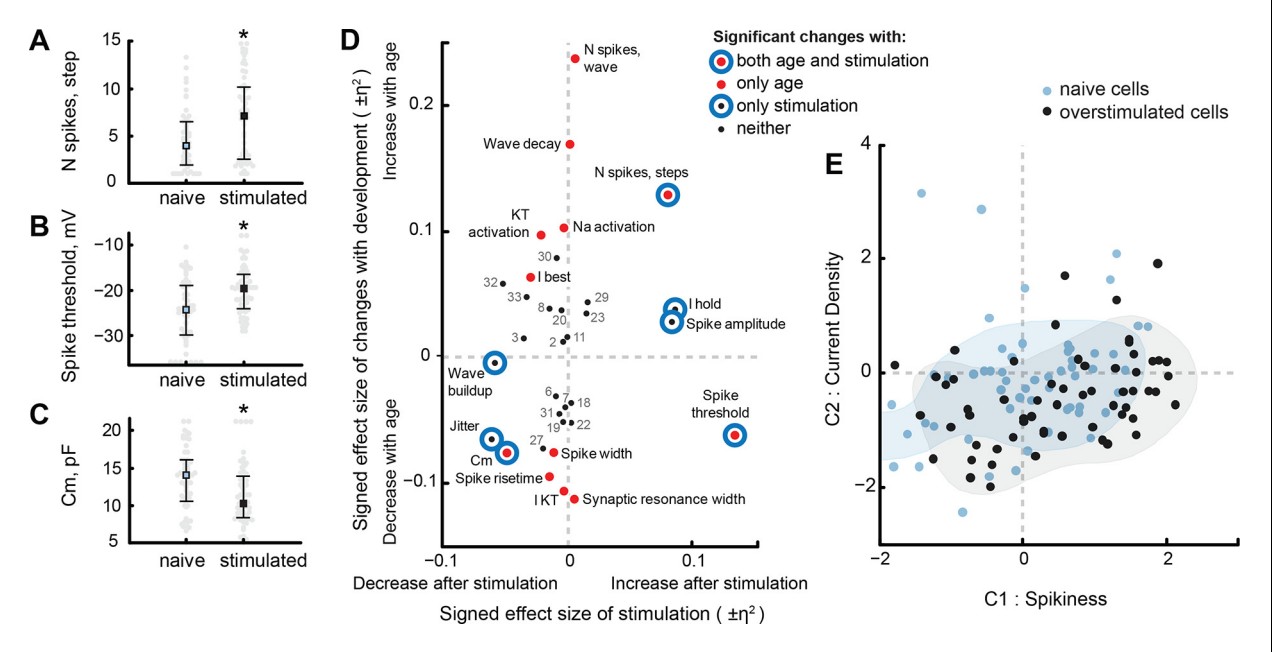

**Figure 7.** Visual stimulation changes some cell properties. (A–C) Visual stimulation increased the spikiness of stage 49 cells in response to step current injections (A), increased spike threshold potential (B), and decreased membrane capacitance of tectal cells (C). (D) A survey of cell properties that significantly changed either during development (red filled markers), in response to visual stimulation (blue hollow circles), or to neither variable (small black markers). Properties that changed both in development and after stimulation are shown as **red** markers with blue circles surrounding them. The position of each marker on the plot is defined by the share of variability explained by developmental stage or visual stimulation, presented as $\eta^2$ effect size value, and taken with a sign that reflects the direction of the change. Properties that did not change significantly are labeled by their number (see 'Materials and methods' or *Supplementary file 3* for the full list). (E) Projection of cells from visually stimulated s49 animals (black) into PCA space defined by the analysis of naïve dataset, with naïve cells from s48-49 animals shown in blue. Shading shows estimated density kernels for respective groups.

of these, one (Spike threshold) changed in opposite directions, and two others changed in the same direction. These results suggest that while visually stimulated cells from stage 49 animals seem to spike similarly to naïve stage 47 animals, the biophysical mechanisms that underlie high cellular excitability in these groups are not likely to be shared.

Curiously, the only electrophysiological property that was significantly affected by both developmental stage and sensory history of the animal, and that could conceivably contribute to differences in cell excitability, was membrane capacitance — a variable that is usually interpreted as an estimation of cell size, and thus does not make an obvious first candidate as a underlying mechanism for short-term intrinsic cellular plasticity. Significant changes in membrane capacitance (*Figure 7D*, *low left corner*) and spike threshold potential (*Figure 7D*, *low right corner*) may indicate a change in electrical coupling between cell soma, from which recordings were performed, and the spike-generating axon hillocks, similar to what has been described in other preparations (*Grubb and Burrone, 2010*; *Kuba et al., 2010*).

To better compare changes after visual stimulation to those occurring over development, we projected electrophysiological data from sensory stimulated cells into the original PCA space defined by data from naïve tadpoles, and compared the resulting point cloud to points from naïve stage 48–49 animals (*Figure 7E*). Compared to the naïve group (*Figure 7E*, blue) the 'Spikiness' (C1) of neurons from visually stimulated animals (*Figure 7E*, black) was higher ($P_{MW} = 0.02$, N = 56 and 60 for naïve and visually stimulated groups), while average 'Current density' (C2 value) of stimulated cells did not change ($P_{MW} = 0.06$). The variances of C1 and C2 did not change significantly ($P_V > 0.05$), however the size of the cloud, as measured by median pairwise Euclidean distance between points in PCA space, decreased in stimulated cells ($P_{MW} < 1e{-}8$). Likewise, the average pairwise difference between cells in the original 33-dimensional space decreased by 11% after visual stimulation ($29 \pm 1$ for stimulated, compared to $33 \pm 1$ for naïve stage 48–49 cells; $P_{TT} < 1e{-}15$). This change in cloud

size is almost certainly because 8 out of 33 electrophysiological properties became significantly less variable after visual stimulation: namely, maximal $Na^+$ and slow $K^+$ voltage-gated currents, transient $K^+$ current activation potential, spiking threshold, jitter, frequency and amplitude of miniature EPSCs, and synaptic resonance inter-stimulus interval ($P_V < 0.05$ for each of these comparisons). Only two properties were more variable in visually stimulated cells: mean number of spikes in response to step injections and membrane resistance Rm ($P_V < 0.01$ for both).

We then analyzed the internal structure of naïve and stimulated datasets from stage 48–49 animals in the same way as it was done for different developmental stages. We found that the amount of internal heterogeneity, as quantified by the agglomerative nesting coefficient, was reduced in stimulated neurons (0.62 ± 0.02) compared to naïve cells (0.67 ± 0.04; $P_{TT} < 1e-15$; *Figure 6A*), while the share of total variance explained by first two components of local PCA was higher in stimulated cells (0.41 ± 0.02) than in naïve cells (0.35 ± 0.04; $P_{TT} < 1e-15$; *Figure 6B*). Together these findings suggest that visual stimulation of tectal neurons increased their propensity to spike and lowered overall cell-to-cell variability and within-group heterogeneity, making cells more alike. Visual stimulation also made different cell properties more interdependent and predictable, consistent with the consequences of a strong homeostatic constraint experienced by these neurons.

## Convergence of spiking phenotypes

Our data suggest that spiking properties of tectal neurons can be modulated in several different ways during development and in response to sensory stimulation. We used our dataset to try to infer biological processes that could underlie adjustment of spike output. To deduce these processes, we looked for basic intrinsic properties that could serve as good predictors of cell spiking output across all experimental groups (220 cells). We ran a set of competitive sequential linear regression models (*Calcagno and de Mazancourt, 2010*) to test whether we could predict the number of spikes produced by cells in response to current step injections based on their low-level electrophysiological properties, namely: membrane resistance (Rm), membrane capacitance (Cm), and both ionic current activation potentials and maximal amplitudes for sodium ($I_{Na}$), stable potassium ($I_{KS}$) and transient potassium ($I_{KT}$) voltage-gated currents (8 variables total). We found that across all cells, the pair of properties that explained most of spike-output variance was a pair of activation potentials for sodium and stable potassium voltage-gated currents ($I_{Na}$ and $I_{KS}$; 18% of variance without interaction, 19% with interaction). After activation potentials were considered, the next most informative variable was membrane resistance (Rm, 7% of variance; 13% with interactions) followed by voltage-gated sodium channel amplitude (2% of explained variance, 4% with interactions), and membrane capacitance (5% without, 15% with interactions). Together these 5 variables explained 33% of variation in the number of spikes if taken without interactions, and as much as 51% if multiple-order interactions were included. Unfortunately, linear models did not help to differentiate between variables underlying tuning of spike output during development and after sensory stimulation, as none of the effects disappeared (based on $P_F < 0.05$ criterion) when both developmental stage and experimental manipulation were factored in.

At the same time, the relative abundance of significant interactions (7 out of 26 investigated in the model) and the high share of variance in spiking output explained by these interactions (18%, compared to main linear effects of 33%) indicated that simple electrophysiological properties included in our analysis do not regulate cell spiking independently, but are likely to be constrained (*O'Leary et al., 2013*). For example, the cell property that predicted most of spiking output variance, voltage-gated sodium current activation potential, interacted significantly ($P_F < 0.05$) with slow potassium current activation potential, membrane resistance and capacitance, as well as several second and third-order combinations of these variables. In practical terms it means that to keep the spike output of a tectal cell constant, a change in any of these variables should be accompanied by a balancing correction of sodium current activation potential, and vice versa. To further investigate this point, we analyzed distributions of low-level electrophysiological properties in a subset of cells that had similar spiking output in response to current injections (*Figure 8*).

To objectively identify cells with similar spike outputs, we combined each cell's responses to consecutive step injections of increasing current amplitudes into one trace and extracted spike-timing data from this trace. We then applied a commonly-used standard cost-based metric sensitive to both number of spikes and their timing to quantify similarity between spike-trains of different cells (*Victor and Purpura, 1996*, *1997*), and used multidimensional scaling to represent a matrix of

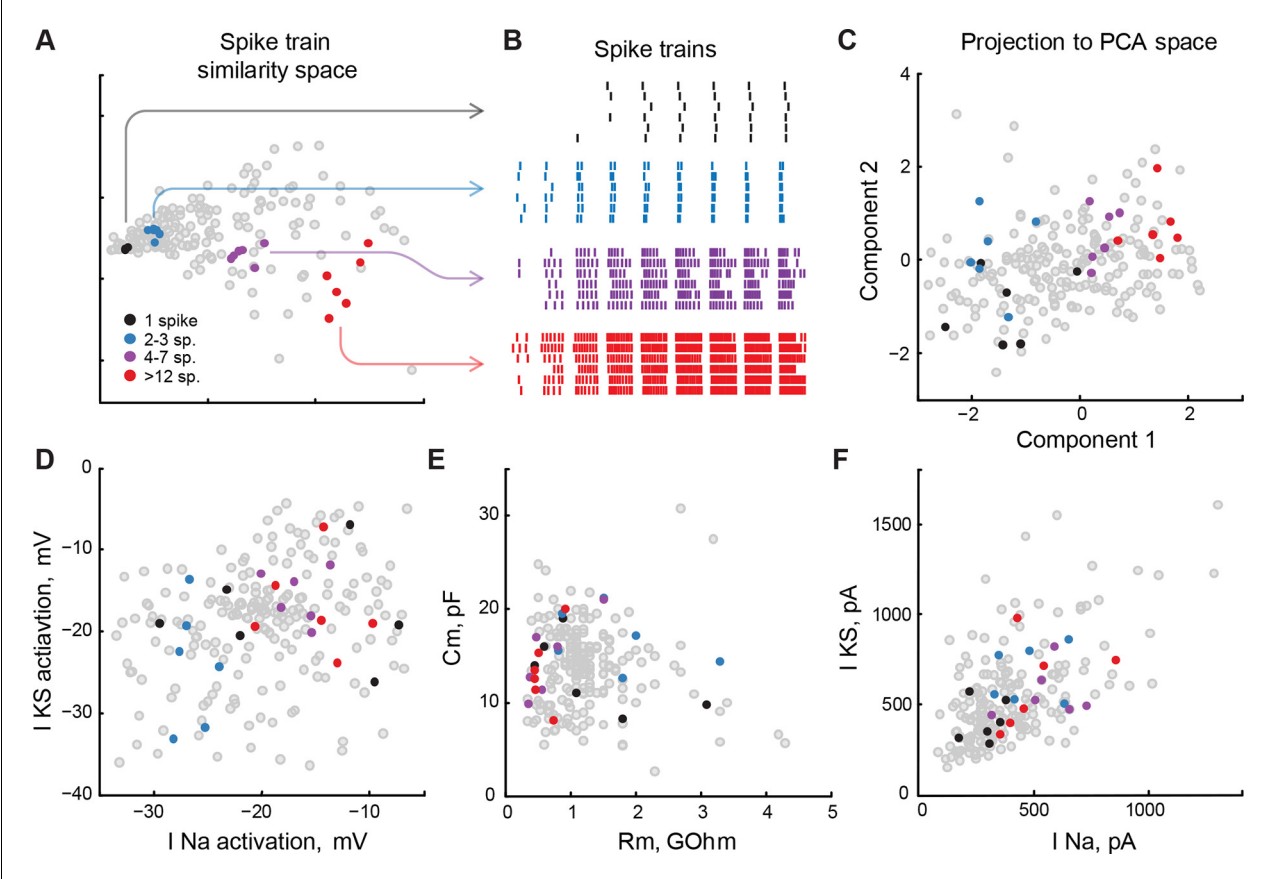

**Figure 8.** Low-level cell properties are a bad predictor for spiking output. (**A**) Multidimensional scaling of differences between cell spiking outputs onto a 2D plane. Cells that produced similar trains of spikes in response to step current injections, both in terms of the total number of spikes, input-output curve, and spike latency, are located nearby. (**B**) Spike-raster for several subsets of 6 cells each shown in panel **A**. Spiking outputs of cells are very different between the groups, but are closely matched within each group. (**C**) Groups of cells from panels **A** and **B**, projected into PCA space that describes the full variability of cell properties. Clusters of cells are still visible, but they are no longer compact, and groups are partially overlapping. (**D**–**F**) The same groups of cells are shown on correlation plots for meaningful (both biologically and statistically; see text) pairs of cell properties: threshold potentials for voltage-gated sodium and stable potassium currents (**D**), membrane resistance and capacitance (**E**), and ionic currents amplitudes (**F**). The clusters are strongly overlapping, suggesting that cells in which a small subset of properties match can be tuned to produce strikingly different spiking outputs. Threshold data in panel **D** is renormalized to avoid overplotting (see 'Materials and methods').

pairwise cell-to-cell distances in a 2D plot (*Figure 8A*). We used this analysis to select groups of cells (6 in each group) that generated similar spike trains in response to current step injections (*Figure 8B*) by pulling 5 nearest neighbors to arbitrarily selected reference cells. Although the spike-trains produced by cells within each group were very similar to each other, we found that these cells did not form compact clusters when projected on the C1-C2 PCA space, but resembled sparse constellations that were partially overlapping between groups (*Figure 8C*). These results suggest that although spiking output was similar between these cells, they differed drastically in other electrophysiological aspects.

Finally, we combined these two approaches and labeled groups of similarly-spiking cells in correlograms of electrophysiological values that were found to be best linear predictors for cell spiking output, as described above. These variables included activation potentials for sodium and slow potassium voltage-gated ionic currents (*Figure 8D*); passive properties, such as membrane resistance and capacitance (*Figure 8E*), and ionic current amplitudes (*Figure 8F*). Although spiking responses of cells (*Figure 8B*) were similar to each other within each group, and strikingly different between the groups, corresponding markers formed neither clusters nor layered structures indicative of low-dimensional constraints that could link different properties together (*Figure 8D,E,F*). This

suggests that in our system, cells with similar spiking phenotypes may have very diverse underlying electrophysiological properties, and conversely, cells that are strikingly different in their spiking output can have very similar low-level physiological properties (*Figure 8*, black and **red** points respectively).

## Discussion

In this study we systematically assessed cell-to-cell electrophysiological variability of primary neurons in the optic tectum of *Xenopus* tadpoles across several developmental periods and in response to sensory stimulation. Our results indicate that during development cells in the deep layer of the tectum become more diverse — although at the stages we studied they do not split into distinct non-overlapping cell types that are reported in the tecta of other species and at later stages of development in frogs (*Lazar, 1973*; *Ewert, 1974*; *Grüsser and Grüsser-Cornehls, 1976*; *Frost and Sun, 2004*; *Kang and Li, 2010*; *Nakagawa and Hongjian, 2010*; *Liu et al., 2011*). We also found that several key electrophysiological properties of tectal cells change over development. We confirmed previously described changes in the average intrinsic excitability of tectal cells with age (*Pratt and Aizenman, 2007*), and showed that at these stages most physiological differences between cells are linked to their overall spikiness (based on the results of Principal Variable Analysis, Principal Component Analysis, and the comparison of statistical efficiency of different protocols).

More importantly, we report an increased diversification of cell phenotypes at later developmental stages, and a shrinkage of this diversity in response to strong sensory stimulation. The cell-to-cell variability remained relatively low at stages 43–47, and different electrophysiological parameters were more random with respect to each other, both in terms of clustering and linear interdependencies between different variables. By stages 48–49 cell variability in the tectum increased, and some internal structure in the PCA cloud began to emerge, with patterns of cell properties agglomerating into clusters, which although poorly resolved at the PCA plot, were noticeable through the quantitative clustering analysis. Complementing previously described receptive field refinement (*Dong et al., 2009*), and temporal decorrelation of spiking activity (*Xu et al., 2011*), this tuning and differentiation of cell properties likely reflects maturation of tectal networks. An increase in cell tuning variability is reminiscent of reports from other experimental models, including mammalian sensory cortex (*Jadhav et al., 2009*; *Yassin et al., 2010*), where the non-uniformity of neuronal recruitment thresholds was shown to be a common feature of developed, functional networks (*Elstrott et al., 2014*).

This emerging structure and differentiation of cell properties was, however, decreased by strongly patterned visual stimulation, which reduced cell-to-cell variability, making neurons more similar to each other electrophysiologically. At the same time, the amount of variance explained by linear correlations between different variables increased after visual stimulation. This suggests that sensory stimulation, and associated homeostatic plasticity (*Aizenman et al., 2003*; *Dong et al., 2009*) left a predictable trace in the mutual arrangement of different physiological properties within each cell (*Turrigiano et al., 1994*; *Dong et al., 2009*; *Munz et al., 2014*). These predictable traces and correlations were then picked up by local factor analysis, making our results similar to reports from the stomatogastric ganglion model (*O'Leary et al., 2013*). We also show that the shift in neuronal excitability induced by visual stimulation was supported by different underlying electrophysiological properties than were the similar changes in excitability observed during development.

Among the practical consequences of this study, we point to developmental stage 47 as a likely candidate for the critical tuning period for tectal network maturation. We describe a previously undocumented sharp, transient increase in excitability in tectal cells during stage 47, providing an explanation for the previously unarticulated practice of aggregating developmental data over stages 45–46 and 48–49, but avoiding pools of stage 47 neurons with other stages (*Pratt et al., 2008*; *Deeg et al., 2009*; *Dong et al., 2009*; *Sharma and Cline, 2010*; *Xu et al., 2011*; *Khakhalin and Aizenman, 2012*; *Spawn and Aizenman, 2012*). This transient development stage, which lasts for only about 12–18 hr, and is traditionally defined solely on the basis of embryonic morphology (*Nieuwkoop and Faber, 1994*), was accompanied by rapid changes in cell tuning variability, and a powerful (almost two-fold) increase in cell excitability. This intriguing developmental pattern could be explored in the future as a model for a critical period in development.

Finally, our analysis of neurons with similar spiking outputs demonstrated that strikingly different combinations of underlying low-level electrophysiological properties can lead to similar spiking phenotypes, and conversely, that the predictability of spiking phenotype from any small set of cell properties is low. This reinforces the notion that the conflicting biological goals of developmental flexibility and stability in response to perturbations rely on the redundancy of parameters underlying dynamic behavior of these systems (*Marder and Taylor, 2011*; *Marder et al., 2014*). The consequence of this redundancy is that multiple parameter configurations can produce phenomenologically identical patterns of network activation (*Goaillard et al., 2009*; *Caplan et al., 2014*).

Altogether, our results provide a promising framework for studying mechanisms of network maturation and calibration in *Xenopus* tadpoles, as well as a unique dataset that will be helpful to inform computational modeling of the optic tectum. In future studies we plan to combine electrophysiological identification of single cells with the transcriptional mapping of relevant genes (*Nelson et al., 2006*; *Schulz et al., 2006*) to further advance our understanding of the molecular biology underlying development and plasticity in dynamic systems.

## Materials and methods

### Animals and housing

Wildtype *Xenopus laevis* adults were bred overnight through natural mating in the Brown University animal care facility. Females were primed with 800U human chorionic gonadotropic (hCG); males were primed with 300U hCG ([1000 U/mL]; Sigma-Aldrich; St. Louis, MO). Embryos were collected the following day; cleaned by removal of unhealthy/unfertilized oocytes, and kept in a variant of 10% Steinberg's Solution (also known as ½x MR) [in mM: 5.8 NaCl, 0.067 KCl, 0.034 Ca(NO$_3$)$_2$ · 4H$_2$O, 0.083 MgSO$_4$ · 7H$_2$O, 5 HEPES; pH 7.4] in incubators at 18–21°C under a 12:12 light:dark cycle. Developmental stages are determined according to Nieuwkoop and Faber (*Nieuwkoop and Faber, 1994*). Under our rearing conditions, tadpoles reach stages 44–46 at 9–12 days post-fertilization (dpf), and 48/9 at 18–20 dpf. Animals between stages 43 and 49 were used in experiments. Tadpoles used to characterize development of tectal electrophysiological properties were taken directly from the 18–21°C incubators, while those stage 49 tadpoles that were used to assess homeostatic changes in the tectum were first placed in a custom black acrylic box with four rows of four green LEDs flashing in sequence at 1 Hz for 4 hr.

Tadpole brains were prepared as described in (*Aizenman et al., 2003*). All experiments were performed between ZT 3–9 (10:00–16:00 EST), where ZT 0 is lights-on for a diurnal animal. In brief, tadpoles were anesthetized with 0.02% (w/v) tricaine methanosulfonate (MS-222) in 10% Steinberg's solution and brains were then dissected out in HEPES-buffered extracellular media (containing in mM: 115 NaCl, 4 KCl, 3 CaCl$_2$, 3 MgCl$_2$, 5 HEPES, 10 glucose, 10 µM glycine; pH 7.2 at 255 mOsm/Kg). To access the soma layer of the tectum, brains were filleted along the dorsal midline and extracted for pinning to a submerged block of Sylgard 184 Silicone Elastomer (Dow Corning; Midland, MI) in a custom recording chamber at room temperature (23°C). Using a large-bore glass electrode, the ventricular membrane was suctioned to reveal the tectal cell body layer.

### Electrophysiology

Tectal cells were visualized using a Nikon (Tokyo, Japan) FN1 light microscope with a 60x water-immersion objective. While a visually heterogeneous population of tectal neurons were selected, care was taken to only patch those principal tectal neurons that looked healthy (clear, no granulation) and to avoid particularly large cells (size and shape) that might be mesencephalic trigeminal neurons (*Pratt and Aizenman, 2009*).

To ensure valid comparisons across stages of development, we restricted our recordings to the middle third of the tectum, thus reducing developmental variability along the rostro-caudal axis (*Wu et al., 1996*; *Khakhalin and Aizenman, 2012*; *Hamodi and Pratt, 2014*). All cells were recorded within 3 hr of dissection. Drugs and chemicals were obtained from Sigma (Sigma-Aldrich; St. Louis, MO).

Glass electrodes were pulled on a Sutter P97 or P1000 puller (Sutter Instruments; Novato, CA) from either Corning 7056 thin wall capillary glass tubing (G75165T-4, Warner Instruments; Hamden, CT) or Sutter thick wall capillary glass tubing (B150-86-10) to a tip resistance of 8–12 MΩ. The

electrodes were then filled with a K-gluconate-based intracellular saline (containing in mM: 100 K-gluconate, 5 NaCl, 8 KCl, 1.5 MgCl$_2$, 20 HEPES, 10 EGTA, 2 ATP, 0.3 GTP; pH 7.2 at 255 mOsm/Kg) through a 200 nm syringe filter (#171, Nalgene-Thermo; Waltham, MA). Filled electrodes were placed in an Axon headstage containing a AgCl wire and controlled by a motorized micromanipulator (MX7600R and MC1000e-R, Siskiyou; Grants Pass, OR). Electrode was lowered into the chamber with slight positive pressure in the pipette until the target cell was contacted, at which time negative pressure was applied to form a high resistance seal (>1GΩ), and then again to break through the neuronal membrane. Whole cell patch clamp electrophysiological signals were measured with an Axon Instruments MultiClamp 700B amplifier, filtered with a 5 kHz band-pass filter, and digitized at 10 kHz by an Axon Instruments DigiData 1440A, and acquired with pCLAMP 10 software (Molecular Devices; Sunnyvale, CA).

Upon initial whole cell patch, a seal test and membrane test measured the following parameters: cell membrane capacitance (Cm), cell membrane resistance (Rm), access resistance (Ra), and holding current (I hold) necessary to keep the membrane at -65 mV. Each clamped neuron was subjected to a series of voltage clamp and current clamp protocols to assess its characteristics. First, cells were held at -65 mV in voltage clamp to ensure that Na$^+$ channels are fully de-inactivated. Then cells were stepped for 150ms to membrane potentials from -65 mV to 115 mV in 20 mV increments to get the data for IV curves. Cells were then switched to current clamp and subjected to ten square pulses of current from 0 pA to 180 pA in 20 pA increments. We next explored potential resonances in tectal neurons (*Hutcheon and Yarom, 2000*) by probing their ability to fire in response to cosine current injections of varying frequencies. Each sweep contained five separate 200 ms bouts of cosine injections with frequencies of 100, 50, 30, 25, and 20 Hz, and peak amplitude of 135 pA. To determine health of the neurons, they were switched back to voltage clamp and subjected to the IV-curve voltage step protocol once again. After that, the spontaneous activity was continuously recorded for one minute at −45 mV holding potential. Continuing to hold at -45mV, cells were then subjected to synaptic stimulation protocols, in which the optic chiasm (OCh) was stimulated with a bipolar stimulating electrode (FHC, Bowdoin, ME) to activate retinal ganglion cell axons. Five stimuli of 150 μA to 800 μA and duration of 180 μs were provided at varying frequencies, with inter-stimulus intervals of 10, 20, 30, 40, 50, 100, 150, 200, 250, and 300 ms; the protocol was repeated 5 times. Finally, the IV-curve voltage step protocol was repeated a third time to ensure the health of the cell and stability of the data. The microscope was then switched to the 10x objective, and the cell location was recorded (*Khakhalin and Aizenman, 2012*). The cell was suctioned away from the tissue and the process was repeated for up to 6 neurons. Some cells did not survive the entire series of protocols, but as long as the nearest IV-curve recording from these cells was stable, they were included in the dataset. No potentials reported in this paper were corrected for the expected junction potential of +12 mV.

## Data processing and analysis

Here we introduce and enumerate variables that were included in the analysis, and are presented further. In total, up to 33 different variables were measured for each cell, with 4 variables coming from a standard seal test; 6 variables from the IV-curve protocol; 10 variables from the step current injection protocol; 6 from the protocol of cosine-shaped current injections of different frequencies; 5 from the synaptic stimulation protocol, and 2 from the recording of spontaneous postsynaptic potentials. Data analysis was performed in MATLAB (Mathworks, MA) and R Studio.

### Passive electric parameters

Based on the standard seal test, as implemented in pClamp 10, cell capacitance (variable #1: Cm, pF); membrane resistance (variable #2: Rm, GΩ), and access resistance (#3: Ra, MΩ) were measured. We did not measure the resting membrane potential Em of each cell, but recorded the holding current (#4: I hold, pA) required to keep the cell membrane at −65 mV. This value is linked to Em by a simple linear equation: I hold = (−65 − Em)/Rm, and thus can be used as a substitute for Em in exploratory analysis. Based on our data, Em in naïve cells was −49 ± 13 and did not change over development (P$_{ANOVA}$>0.05); in stimulated cells, Em increased to −35 ± 27, which was significant (P$_{ANOVA}$ = 1e−3), and translated from significant changes in I hold (see Results). It is also important to note that tectal cells are relatively small, and in them the value of Em is dominated by the ionic

concentrations of the internal and external solutions. The original resting membrane potential is disrupted within 2–3 seconds after the whole-cell patch is established (*Khakhalin and Aizenman, 2012*).

## IV protocol

It was previously shown (*Aizenman et al., 2003*) that in *Xenopus* tadpoles OT neurons Na$^+$ and K$^+$ ionic currents are isolated enough temporally to allow their simultaneous measurements from traces produced by brief membrane depolarization in voltage-clamp mode (*Figure 1A*). A time window from 0 to 185ms after the membrane potential change was used to estimate peak sodium (I$_{Na}$) and potassium currents (*Figure 1A*, *right*); an average current over the window from 1165 to 1365ms after the potential change was used as an estimation of the steady state potassium current (I$_{KS}$; *Figure 1A*, *left*). The amplitude of transient potassium current I$_{KT}$ was estimated as a difference between peak potassium current within the first window and the steady state potassium current. To quantify IV curves for these currents, for each cell the curves were fit with a smooth function of membrane potential, and the parameters of this fit function were reported. For I$_{Na}$ and I$_{KT}$ currents (*Figure 1B,D*) fit curves were defined as

$$I(v) = c/1 + exp(-(v-a)/b + d)$$

where *v* is the depolarization step potential and *a-d* are parameters. The potential at which each of these ionic currents reached ½ of its maximal value, and the maximal current (maximum of the fit curve) were used as variables #5 (I$_{Na}$ activation, mV); #6 (I$_{Na}$, pA); #9 (I$_{KT}$ activation, mV), and #10 (I$_{KT}$, pA). For I$_{KS}$ current IV curve (*Figure 1C*) was fit with a model

$$I(v) = max(0, exp((v-a)/b) - e) \cdot c + d$$

where *v* is the potential, and *a-d* are parameters. The first potential at which I$_{Ks}$ was activated and the fit curve maximum were reported as variables #7 (I$_{KS}$ activation, mV), and #8 (I$_{KS}$, pA). Note that 'activation potentials' are empirical potentials at which macroscopic ionic currents were activated, and they are likely to be different from threshold potentials of individual channels, both because of a different mathematical definition of these potentials, and because macroscopic activation potentials are also affected by the geometry and cable characteristics of each tectal cell.

As in IV-curve experiments we tested holding potentials at increments of 10 mV, and as activation of voltage-gated currents was sharp, the distribution of ionic current activation potential estimations had modes around discrete values separated by 10 mV. Raw data was used for all calculations, but in *Figure 8D*, to avoid overplotting, we homeomorphically transformed the data by first rank-transforming it, and then scaling it back from ranks to original readings in mV using a least squares best fit cubic polynomial. This mapping strictly preserved the relative arrangement of points, but made the local density of points in *Figure 8D* less banded.

## Step current injection protocol

For each step current injection (*Figure 1E*), the number of evoked spikes, their amplitudes, and latencies at peak were measured. Spikes were detected automatically through adaptive filtering with subsequent thresholding, which discriminated against spikelet shapes that were either too small or too broad. All results of spike detection were also visually verified by two people blinded to neuronal identity. The amplitude of each spike was measured as a difference between peak potential during the spike and the potential at the kink point, defined as a point at which the 2nd derivative of potential over time went through a maximum (*Figure 1F*). For every spike, rise time (10% to 90% of potential increase from kink point to the peak) and width at half-height (measured at potential between the kink point and the peak) were measured automatically (*Figure 1F*). As the current injection ended, and the neuron repolarized, the shape of this repolarization potential curve was fit exponentially.

For every cell the following properties were reported: median time constant of repolarization after current injection as variable #11 'Tail' (*Figure 1E*); potential of the kink point of the first spike generated at the smallest current injection as #12 'Spike threshold' (*Figure 1E* , **blue** traces); amplitude of the first spike at the smallest current potential as #13 'Spike amplitude', and its rise time and width at half-length as #14 'Spike rise-time' and #15 'Spike width' respectively (*Figure 1F*).

The number of evoked spikes as a function of injected current (*Figure 1G*) was fitted with a curve:

$$ns(i) = max(0, \, exp(-(i-a)/b) \, - \, exp(-(i-a)/c)) \cdot d$$

where *i* is current, and *a-d* are fit parameters. From this fit two variables were estimated: the current that generated maximal spiking (defined as x-coordinate of fit curve maximum) as #16 'I best', and weighted maximal spiking, estimated as maximum of a fit curve, as #17 'N spikes, step'.

For the trace that produced highest number of spikes and was the closest to the inferred 'optimal current' (*Figure 1E*, black trace), and if more than one spike was generated, we reported the inter-spike interval in ms (#18, 'Spike ISI'). If more than two spikes were generated we also reported the ratio between the 2nd and the 1st inter-spike intervals (#19, 'Spike ISI accommodation'). For cells that generated at least 2 spikes, the amplitude ratio of the 1st and the 2nd spikes in the train was reported as #20 'Spike accommodation' (*Figure 1E*).

To identify cells that produced similar spike-trains in response to step current injections (as presented in *Figure 8*) we used a standard cost-based metric of spike-train similarity (*Victor and Purpura, 1996*, *1997*) with a cost of 0.1/ms for spike-timing adjustments; this procedure is sensitive to both number of spikes and their latencies. For this calculation, responses to step current injections of all amplitudes were combined and treated as one long recording (similar to that shown in *Figure 1H* , but for step, rather than cosine injections).

## Cosine current injections of different frequencies

To assess dynamic resonances in tectal neurons (*Tan and Borst, 2007*) we injected them with cosine-shaped currents of different frequencies (*Figure 1H*). Spikes were detected in MATLAB and verified visually, similar to step-injections. The number of spikes as a function of wave period (*Figure 1J*) was fit with the formula:

$$n(T) = max[0, \, exp(-[T-a]/b) \, - \, exp(-[T-a]/c)] \cdot d$$

where *T* stands for wave period, while *a-d* are optimization parameters. From this fit, the optimal wave period in ms was estimated (variable #22, 'Spiking resonance'). Mean number of spikes averaged across waves of all frequencies was reported as #21 'N spikes, cosine.' The time constant of spike-output build-up with cosine injection period increase (parameter *b* from the formula above) was reported as a measure of spike-non-inactivation in response to slow-frequency currents (#23: 'Spiking resonance width').

For the highest frequency of cosine current injections (100 Hz) number of spikes in response to each current wave, taken as a function of wave number *ns(x)*, was fit with the formula:

$$ns(x) = (x-a) \cdot exp(-(x-b)/c) \cdot d + e$$

where x is a continuous independent variable interpolating integer wave numbers, and *a-e* are fit parameters (*Figure 1K*). From this fit we inferred properties of spiking activation and inactivation in each cell, and reported the wave number that was expected to produce highest spiking in this set based on the fit (#24, 'Wave buildup'), and the speed of spike number adaptation, given by the *c* parameter from the formula above (#25, 'Wave decay').

Cosine current injections of different frequencies were also used to estimate the index of spiking unpredictability, or jitter (#26, 'Jitter'). To estimate this value, 10 different spike trains generated in response to cosine current injections (*Figure 1I*) were represented by δ-functions in a 10 kHz trace, convolved with a Gaussian ($\sigma$ = 2 ms) and normalized. Pairwise scalar products (correlations) were calculated for each suitable pair of traces; these correlations were then averaged across all sweep pairs. The resulting value represented a measure of spike-time consistency, as it would be equal to 1 for identical trains, and approach 0 for perfectly unmatched trains. To move from a 'consistency index' to a 'jitter index' we inversed the value, and calculated a natural logarithm of it. The final formula could thus be expressed as:

$$Jitter = \ln[1 - mean(scalarProduct(a_i, \, a_j))]$$

where

$$a_j = \text{conv}(\text{spikeTrain}_j, \text{gaussian}(\sigma = 2\,\text{ms}))$$

## Synaptic stimulation protocol

For responses to synaptic stimulation (*Figure 1J*), we removed stimulation artifacts, averaged trials with the same inter-stimulus intervals (ISIs), and calculated total synaptic charge for each of the ISIs. The total charge Q as a function of ISI (*Figure 1M*) was fit with a function:

$$Q(\tau) = (\tau - a) \cdot exp[-(\tau - b)/c] \cdot d + e$$

where τ stands for the ISI value in ms, and *a-e* are fit parameters. From this fit we reported estimated optimal inter-stimulus interval to produce maximal synaptic input as #27 'Synaptic resonance' (ms); the parameter *c* from the fit formula as #28 'Synaptic resonance width' (the measure of sharpness of synaptic frequency tuning), and the maximal total synaptic charge produced in a cell as #29 'Synaptic charge' (pA·s). We also calculated the ratio between the maximal total synaptic charge observed in each cell and the projected total charge in response to infinitely slow stimulation (max Q/e from the fit formula above). We dubbed this variable #30 'Synaptic PPF' and interpreted it as a measure of non-linear synaptic summation, even though unlike actual Paired-Pulse Facilitation (PPF), our measure was based on five, and not 2 stimuli; involved a ratio of total charges, and was reported for the best inter-stimulus interval out of 10 different intervals we tried for each cell.

Finally, traces with longer inter-stimulus intervals (100, 150, 200, 250 and 300 ms) were used to compare the average amplitude of early monosynaptic components of the response (those measured between 5 and 14 ms after the shock at the optic chiasm) to that of polysynaptic recurrent responses that occurred later after the stimulus (observed from 15 to 145 ms after the shock; *Figure 1N*). The ratio of average currents over these time windows was interpreted as the measurement of relative strength of monosynaptic and polysynaptic (recurrent) inputs to each cell; it was reported as variable #31, 'Monosynapticity'.

## Spontaneous synaptic events

For 64 cells we recorded spontaneous synaptic activity, and calculated average frequency (#32, 'Minis frequency') and amplitude (#33, 'Minis amplitude') of spontaneous excitatory postsynaptic currents. While spiking activity in the preparation was not blocked, and so some of these spontaneous events could have been influenced by background spiking in the network, our previously published results show that these data can be used as good proxy for 'true' miniature postsynaptic potentials (*Pratt and Aizenman, 2007*).

## Potential sources of variability

To get a better understanding of sources of cell properties variability, we checked whether values of any of 33 variables we measured correlated with cell location within the tectum, as previously described in (*Wu et al., 1996*; *Khakhalin and Aizenman, 2012*; *Hamodi and Pratt, 2014*), and despite our attempts to consistently sample from the middle third of the tectum. For tadpoles at stages 48–49, from the main dataset (see below), only two variables significantly correlated with rostro-caudal distance in our preparation (after FDR adjustment of $P_{corr}$ with α = 0.05): membrane capacitance Cm (r = −0.41, N = 46), and membrane resistance Rm (r = −0.42, N = 46). While comparison of rostro-caudal distances between different developmental stages is problematic, there was no difference in rostro-caudal distances of sampled cells between main (naïve) and visually stimulated experimental sets at stages 48–49 ($P_{MW}$ = 0.3, N = 20 and 36 respectively). We also looked at whether the age of the preparation (time since dissection in hours) and the time of the day (assuming potential presence of circadian or diurnal rhythmic modulation in the tectum) affected any of our variables (based on significant correlation after FDR adjustment with α = 0.05). For naïve tadpoles at stages 48–49 the preparation age did not affect any of the variables we measured, and none of the variables were affected by the time of day at which cells were recorded.

## Statistical procedures

As many variables analyzed in this paper were not normally distributed, and to stay consistent, we preferred Mann-Whitney two-sample tests for comparing data between groups; p-values of this test

were reported as $P_{MW}$. At the same time, to make referencing and subsequent meta-analysis possible, we always report means and standard deviations in the text. Other statistical tests used in this study include Pearson correlation ($P_{corr}$), one-way ANOVA ($P_{ANOVA}$), multiple linear regression test ($P_F$), Welch generalization of Student's t-test ($P_{TT}$), and F-test for equality of variance ($P_V$). When exploring potential correlations between variables we used False Discovery Rate (FDR) procedure to adjust for multiple testing, keeping the false discovery rate at 0.05 (*Benjamini and Hochberg, 1995*). The FDR procedure was performed separately for each massive set of comparisons, as indicated in the text. In post-hoc pairwise comparisons of data groups after ANOVA tests we used a more conservative Bonferroni correction, but reported uncorrected p-values. To illustrate correlation levels between variables (*Figure 2A*) we used a custom MATLAB script inspired by the circular diagram from the 'Circos' visualization package (*Krzywinski et al., 2009*). As not every pair of variables was available in every cell, different correlation tests were run on different numbers of points (from 38 to 154; median of 108). We verified our Pearson correlation-based analysis by calculating Spearman correlation coefficients; after FDR 112 Spearman correlations were significant (as opposed to 90 for Pearson), and all Pearson correlations with r>0.5 (n=11), including all shown in *Figure 2B*, remained significant for Spearman calculation.

We ran the Principal Variables Analysis in R — package 'subselect' (*Mccabe, 1984*; *Cadima and Jolliffe, 2001*; *Cadima et al., 2004*) — using the correlation matrix of original non-imputed data, which was computed on all available pairwise observations for each pair of variables. We then compared and presented squared RM coefficients (*Cadima and Jolliffe, 2001*), to make the results of this analysis comparable to that of factor analysis below.

As not all measurements were available for every cell in the dataset, for factor analysis we used a generalization of a standard Principal Component Analysis (PCA) procedure, called the Variational Bayesian PCA, or the PCA with missing values (PCA-MV; (*Ilin and Raiko, 2010*); MATLAB m-files are available at the web-site of Bayesian Group, Aalto University, Finland). To verify that the PCA-MV procedure is applicable to our data, we selected a subset of 52 cells and 27 variables to form a full matrix free of missing values. A standard PCA was then run on this reduced data set, and the results of it were compared to the results of PCA-MV run on the full data set. The scree plot of a standard PCA on restricted data suggested that two first components (explaining 20% and 16% of variation respectively) could be interpreted meaningfully; the remaining components representing individual variability of the data, or 'noise'; see (*Shabalin and Nobel, 2013*) for references. We therefore only present the first two PCA-MV components in this paper (explaining 15% and 8% of total variance respectively). Component scores found by standard PCA and PCA-MV on the same subset of cells (N = 52) were highly correlated (r = 0.90 and 0.85 for components 1 and 2 respectively; p < 3e−15), suggesting that the PCA-MV is indeed applicable to our data set.

As 18% of all possible measurements in our dateset were missing, the total share of explained variance (23%) was almost certainly underestimated, and cannot be compared to variance explained by standard PCA. This statement is obvious if you consider that the estimation of total variance in a set with randomly missing values is unbiased, yet explained variance is biased, as missing values cannot contribute to the calculation: while predictions for them are available, the values themselves are not present. As described above, a standard PCA run on a subset of data with full representation of all cells and variables explained 36% of total variance. Similarly, when we performed multiple imputation of missing values using R package 'Mi' (*Su et al., 2011*) (see below for details), a standard PCA procedure explained on average 35 ± 5% of total variance.

Some of the variables we assessed were distributed non-normally, and we attempted to run PCA-MV on renormalized rank-transformed variables, and compared the results of this analysis to that of PCA-MV run on raw variables. Rank-based normalization improved the amount of variance explained by the first component (from 15% on raw data to 21% on rank-transformed data), but did not improve explanatory value of higher components. Upon visual comparison of score-plots and loading-plots, we concluded that the relative arrangement of individual cells (score-plot), as well as contributing variables within the 2D plane of first two components (loading-plot), did not change enough to justify the use of rank-transformation. All analysis reported in the paper was therefore performed on raw variables.

While linear approaches to factor analysis, such as PCA or Multidimensional Scaling, are usually considered to be safe and preferable methods when noisy and weakly correlated data are concerned (*Nowak et al., 2003*; *Sobie, 2009*; *McGarry et al., 2010*), we compared the performance of PCA to

the two most popular non-linear 2D ordination approaches: Isomap and Local Linear Embedding. To quantify the quality of 2D ordination we looked at how well the 2D map preserved pairwise differences between points in the original 33D space, using the squared correlation coefficient $R^2$ between 2D and 33D distances as an output measure (*Pedhazur, 1982*). Based on this metric, PCA preserved 51 ± 2% of variance in pairwise differences (5 alternative Bayesian imputations of missing data using R package 'Mi'). The quality of Isomap projection improved as the projection became less and less local, from 17 ± 4% for isomap based on 3 closest neighbors for each point, to 36 ± 7% based on 20 closest neighbors; still it was substantially lower than for PCA. The Local Linear Embedding approach (R package 'lle', based on (*Kouropteva et al., 2005*) also produced better results as more neighbors were considered, with the local best solution achieved at 19 neighbors explaining 27 ± 9% of variance in pairwise differences (as opposed to 51 ± 2% for PCA). Based on these results we concluded that for our data linear factor analysis approach is not only adequate, but also the most appropriate. In all cases testing was performed on centered and normalized data.

We also attempted restricting the number of variables included in PCA by pre-screening them based on their Principal Variables rank and leaving only variables that explained high amounts of total variance in the dataset. At 17 variables (one half of the original set, corresponding to total explained variance threshold of 6%) PCA-MV explained 40% of total variance in the set (as opposed to 23% for full data PCA-MV), and some of the effects we describe in the paper became more prominent (for example, F-value for changes in PCA cloud size across developmental stages increased from 75 for the full set to 118 for restricted set). However we decided not to present PCA of restricted data in the paper, as thinning out of the multivariate dataset is generally not recommended for exploratory analysis when there is no objective post-hoc test to justify the use of one restricted model over another (*Guyon and Elisseeff, 2003*). We therefore only report it here as another validation of the method.

To simplify interpretation of loading- and score-plots we performed 'promax' oblique rotation of first two PCA-MV components using a standard 'rotatefactors' routine from MATLAB statistics toolbox. This approach maximizes varimax criterion using orthogonal rotation, and further simplifies the projection by applying Procrustes oblique rotation, using orthogonal rotation from the first step as a target.

To compare cloud sizes in the PCA space, we calculated all possible pairwise 2D Euclidian distances between the cells and compared their medians. To illustrate positions, shapes and spreads of clusters at the PCA scores plot in *Figure 5* and *Figure 7* we used the Kernel Density Estimation procedure (*Botev et al., 2010*). To compare cluster sizes in the original 33-dimensional space, we centered and normalized the data, performed 50 alternative Bayesian imputations of missing values using R package 'Mi' (*Su et al., 2011*), and for each imputation subsampled 10 different sets of 50 points to compensate for small differences in original dataset sizes (n = 64, 56 and 60 for naïve stage 44–45, naïve stage 48–49 and visually stimulated stage 48–49 respectively). For each sampled subset of points we computed city-block ("Manhattan") pairwise distances between all cells in the subset, and calculated the median of these values. Finally, we used a t-test to compare sets of medians between data groups (resulting in n = 500 values for each group).

To quantify the amount of internal heterogeneity within data sets, we ran agglomerative nesting analysis (AGNES from package 'cluster') in R, and used the agglomerative nesting coefficient as a measure of heterogeneity (*Struyf et al., 1996*). In practice, we used the same imputation procedure as described above (50 imputations, each contributing to 10 subsets of 50 points each), applied the AGNES clustering procedure to these data, saved agglomerative clustering coefficients, and finally compared them across data groups. Similarly, for 'local within-group PCA' we used same imputation / subsetting procedure, ran PCA on each set, and calculated the share of variance explained by first two components.

To statistically link the number of spikes produced in response to step injections to basic electrophysiological properties of each cell we built a family of general linear models using non-marginal sequential GLM statistics in R package 'glmulti' (*Calcagno and de Mazancourt, 2010*).

## Acknowledgements

The authors would like to thank Irina Sears and Phouangmaly Mimi Oupravanh for animal and lab care; Aaron Wetzler (Israel Institute of Technology) for sharing useful MATLAB scripts; Andrey

Shabalin (Virginia Commonwealth University) for his advice on statistical methods used in this paper. This work was supported by T32 MH019118-19 (B. Connors for CMC), the Sidney A. Fox and Dorothea Doctors Fox Postdoctoral Fellowships for Vision Research (CMC, ASK), and NSF IOS 1353044 (CDA).

## Additional information

### Funding

| Funder | Grant reference number | Author |
|---|---|---|
| National Institutes of Health | Institutional T32 | Christopher M Ciarleglio |
| National Science Foundation | IOS 1353044 | Arseny S Khakhalin Carlos D Aizenman |
| Brown University | Fox postdoctoral fellowship, UTRA Program | Christopher M Ciarleglio Arseny S Khakhalin |

The funders had no role in study design, data collection and interpretation, or the decision to submit the work for publication.

### Author contributions

CMC, Conception and design, Acquisition of data, Analysis and interpretation of data, Drafting or revising the article; ASK, CDA, Conception and design, Analysis and interpretation of data, Drafting or revising the article; AFW, Created and troubleshooted analysis routines to extract parametric data, and contributed to the overall analysis design., Acquisition of data, Analysis and interpretation of data; ACC, Contributed to help develop data analysis tools., Analysis and interpretation of data; SPY, Contributed with data acquisition and experimental design in the very early stages of the project., Acquisition of data

### Author ORCIDs

Arseny S Khakhalin, http://orcid.org/0000-0002-0429-1728

### Ethics

Animal experimentation: All handling of animals was approved by Brown University IACUC in accordance with NIH guidelines. The animal protocol used for these experiments is "Regulation of Neural Excitability and Synaptic Function by Experience in the Developing Visual System (#1308000008C002)".

## Additional files

### Supplementary files

• Supplementary file 1. Table describing the variables measured from each cell. 33 variables were extracted from this dataset, the table describes the name of the variables, units. average value and brief description. For more details about each variable please see Materials and methods.

• Supplementary file 2. Spreadsheet containing data table of parameters extracted from every cell. This spreadsheet contains the extracted parameters from every cell in this study. These data was used for generating principal component analysis and other statistical measures.

• Supplementary file 3. Average values for different stages and experimental conditions. The data in this table describes the average values of each measured variable across several cells grouped by developmental stage and experimental condition.

• Source code 1. Matlab and R scripts used to analyze data. This set of custom functions was not designed for public use, so many aspects of data analysis in these functions are hard-coded (usually as a global variable explicitly defined in the beginning of each script). Note also that at the submission stage we transformed all raw data for this paper from pClamp ABF files to Matlab MAT-file

format, and this is how the data was uploaded to Dryad. The original scripts however worked with our original data files that were mostly stored in either ABF or Microsoft Excel formats.

### Major datasets

The following datasets were generated:

| Author(s) | Year | Dataset title | Dataset URL | Database, license, and accessibility information |
|-----------|------|---------------|-------------|--------------------------------------------------|
| Ciarleglio CM, Kha-khalin AS, Wang A, Constantino A, Yip S, Aizenman CD | 2015 | Data from: Multivariate analysis of electrophysiological diversity of Xenopus visual neurons during development and plasticity | http://dx.doi.org/10.5061/dryad.18kk6 | Available at Dryad Digital Repository under a CC0 Public Domain Dedication |

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
