## [Decision Letter]

[Editors’ note: a previous version of this study was rejected after peer review, but the authors submitted for reconsideration. The first decision letter after peer review is shown below.]

Thank you for choosing to send your work entitled "On the difficulty of predicting neuronal output from underlying electrophysiological parameters" for consideration at *eLife*. Your full submission has been evaluated by Eve Marder (Senior editor) and three peer reviewers, and the decision was reached after discussions between the reviewers. Based on our discussions and the individual reviews below, we regret to inform you that your work will not be considered further for publication in *eLife*.

It has taken us a little time to put together some sort of consensus decision about your manuscript because each person who looks at it seems to see (and not see) somewhat different strengths and weaknesses. I believe that this is a measure of the fact that it is an ambitious undertaking and only partially successful in its present form. Consequently, we are forced to reject this version of the manuscript. Nonetheless, if you find the criticisms, comments here and below helpful, and you feel that you can produce a different version of this work that takes into consideration these reviews, *eLife* would be willing to consider a new version, which we would consider as a new submission. That manuscript might be evaluated by the same or different reviewers, and there is no assurance that it would be successful in review.

Major Strength:

We are all impressed that you have made a serious effort to collect as many measures as possible from each of a large number of neurons in the frog tectum. In principle, this should constitute an important data set, and could be used to answer a number of interesting questions. And one of the reviewers was, and remains positive about the paper, precisely for this reason. Clearly, the attempt to connect the intrinsic properties and electrophysiological behaviors of the neurons to their underlying component processes is deeply important. But, the devil is in the details, and that is where other reviewers had serious reservations, some of which are expressed in the original reviews and others that surfaced during an extensive set of discussions among the reviewers.

Weaknesses/Limitations:

1) The reviewers were unclear about whether your starting assumption was that all of the recorded neurons were of the same cell type: principal neurons of that tectal region? You made it clear that you were excluding neurons that should have been of a different type, but what evidence is there that all of these neurons should be considered similar? In other words, was it your goal to cluster neurons into reliably different subtypes, or was it your goal to study the range of properties in one type of neuron? The manuscript seems to slide back and forth between these two positions.

2) Having 33 attributes sounds impressive, but the question is how many of these attributes are independent, and how many are derived properties? To be more specific, we would all agree that the Na and K conductance densities are independent attributes. But, spike frequency and interspike interval are two measures of the same property. Obviously, this latter case is trivial, but there are instances in which two attributes that on first glance may appear to be independent are not. So for example, membrane capacitance and synaptic current might together give you the envelope of the synaptic events? If so, then is it fair to use all three as independent attributes? It is not clear how many truly independent measurements there are among the 33?

3) We understand that there is an inherent problem when one is trying to measure many properties from the same neurons, and that it may not be possible to make all measurements perfectly. That said, we don't fully understand why particular choices were made. For example, you do not report the resting potential. And we understand why you made some measurements after bringing all the neurons to -65mV. But that by itself was a decision to not measure the voltage difference between the resting potential and threshold. There are a number of choices of this sort which were not adequately justified in the manuscript, but which might change significantly the take-home messages? You report the number of spikes in response to cosine drive, but from -65mV. These data might look totally different if you had started at the neurons' resting potential (which also has its difficulties of course).

4) A relatively minor consideration that however influences how people approach the paper: the title promises more than the paper delivers and sets it up for criticism.

The full individual reviews are below.

Reviewer #1:

In the present article, Ciarleglio and colleagues investigate the changes in electrical phenotype of neurons of the optic tectum of *Xenopus laevis* tadpole occurring during development and after imposed sensory stimulation. One of the specificities of the current article is that the authors use different types of multi-dimensional analyses (mostly PCA) to analyze variations in electrical phenotype defined by 33 intrinsic electrophysiological parameters (passive properties, spiking response, and measurements of specific voltage-dependent ion currents). The main conclusion drawn from this analysis is that a high degree of degeneracy underlies the electrical phenotype of tectal neurons, as neurons with similar spiking can display strikingly different underlying electrophysiological properties and neurons with similar underlying electrophysiological properties may display strikingly different electrical phenotype. Moreover, the authors suggest that the differences in electrophysiological properties found between visually-stimulated and early stages of development for similar spiking profiles also demonstrate degeneracy of electrical phenotype. Although the approach is interesting, I have several concerns about the measurements, the analysis and the conclusions drawn from the results.

1) My main concern is about the poor representativeness of the total variance of the observations by the two principal components used for the PCA (presented in Figure 4, Figure 5, Figure 6, and Figure 7). As the authors mention, the 2 principal components account for 23% of the variance of the observations (15% for PC1 and 8% for PC2). This means that the main part of the variance in the observations (77%) is unaccounted for in this analysis. The choice of analyzing neuronal output by measuring a large number of electrophysiological parameters, and using dimensionality-reduction techniques such as PCA to visualize it is a valid approach if dimensionality reduction preserves most of the information contained in the original measurements. In the present case, 77% of the variance (most likely including variance in parameters that are essential for defining neuronal output) is lost in the analysis. This problem may arise from several different sources: i) the high number of parameters used as an input for the PCA (33) ii) the fact that a linear model such as PCA might not be adequate to describe the variance of the parameters analyzed here. Independent of the source of confusion, this means that PCA may not be the adequate model to use in the present study to draw conclusions on the relationships between neuronal output and underlying parameters. In particular, the concept of degeneracy requires that both the target phenotype (here neuronal output) and the underlying parameters are accurately measured (see main point 2) and defined, and that they have relevant relationships. The fact that PCA accounts for only 23% of the variance in "electrical phenotype" is a real concern that hinders the purpose and the conclusions of this study.

2) My second main concern is about the relevance of some of the electrophysiological parameters measured: some of the parameters are flawed with measurements errors, whilst others lack obvious physiological relevance. The thresholds of the 3 ion currents (Na, KT, and KS) are not measured properly as they are contaminated by reversal potential variations (Na, KT) or are based on a non-sigmoidal fragment of the IV curve (KS). Ion current "thresholds" are usually characterized by precisely defining the half-activation voltages based on conductance vs voltage curves. The measurements of action potential properties in these cells seem to be strongly influenced by variations in the remote location of the spike initiation site (as indicated by the small amplitude of the action potential, around 15–20 mV). In this context, it is difficult to understand whether variations in the kinetics and amplitude of the action potential are mainly attributable to variations in Na and K currents, or variations in the passive properties (i.e. rather related to Cm and Rm) of the compartment located in between the recording site and the spike initiation site. I do not understand the monosynapticity factor.

3) Because of the first two main concerns, it is difficult to really trust the conclusion about the degeneracy of neuronal output in these neurons. Although there is no doubt that biological systems are "degenerate", demonstrating degeneracy first requires to demonstrate that the variable parameters have a strong influence on the output of the system. As an example, action potential shape is degenerate if i) it relies on variable sodium current density and ii) sodium current density has been demonstrated to have a causal influence on action potential shape. The degeneracy argument is irrelevant when the second condition is not fulfilled: as an example, the fact that synaptic transmission at axon terminals located far away from the soma tolerates large variations in soma size does not tell us anything about the degeneracy of synaptic transmission, until we prove that soma size has a significant influence on synaptic transmission.

Reviewer #2:

Ciarleglio, Khakalin et al. have analyzed input-output tuning (spiking) and the underlying electrophysiological properties of neurons in the optic tectum of *Xenopus* tadpoles over a series of stages of development and following visual stimulation at a single stage. Principal component analysis identified properties of spikiness and robustness. Some properties were found to change during development, while others changed with stimulation. Neuronal properties were observed to become more diverse with age. The authors observes substantial degeneracy in that multiple combinations of electrophysiological properties lead to similar spiking behavior so that electrical properties cannot predict spiking activity.

The project is very well conceptualized and executed. The paper is clearly and very well written and ties to the figures nicely. The figures in turn are very well assembled and easy to follow. Figure 7 is particularly informative.

My only suggestion is to change the title to something more positive. Perhaps something like Quantitative analysis of the relation of spiking output to electrophysiological properties reveals substantial degeneracy. I think this will motivate more readers to dip into it.

The authors have addressed an important question about structure (component) – function relationships in the vertebrate brain.

Reviewer #2 (Additional data files and statistical comments (optional)):

The data files and statistical analyses seem appropriate.

Reviewer #3:

The authors have provided a substantial data set of electrophysiological properties of tectal neurons of *Xenopus* tadpoles, towards a goal of better understanding the changes in spiking properties of these neurons over development. An additional, and perhaps more striking, point of the paper is the apparent variability of properties that underlie "similar" spiking patterns. In its current form, I'm not sure it has accomplished these goals. There is also a much richer literature in which to root these ideas, particularly in computational arenas, which are largely overlooked.

There is inherent value in large electrophysiology data sets, and the authors provide a substantial amount of data in this regard. Ultimately this paper is about analysis choices within this data set. I have comments about some of these analysis choices, organized by figures:

Figure 2: The authors seem to have chosen to combine a large dataset of heterogeneous cell types and then perform post-hoc correlation analyses. The rationale for this is unclear, and potentially confounding. The value of the "network" style plot in Figure 2 is minimal. This kind of diagram could potentially be valuable for comparing correlation patterns across groups, but as a standalone, it is difficult to follow any one pair of parameters that the reader would be interested in. The thickness of the lines has no interpretive value without a scale associated. For example, R-values that range from 0.1 to 0.3 would still have a threefold range of thickness, but show minimal correlation regardless.

Figure 5: Using a PCA, despite the difficulties inherent in not all data being present for all cell types, can be a powerful approach. However, while one can somewhat impute characterizing "spikiness", a meaningful definition of "robustness" is not provided. The authors then go on to make the point that the PCA cloud is moving across developmental "space", predominantly in the "spikiness" domain. This is seemingly a bit of an overwrought analysis, when a much more direct measure of "spikiness" (i.e. the spiking of the cells) is provided in Figure 3 and demonstrates in much less convoluted terms that spiking propensity does indeed change in these groups.

Figure 7: This is a major thrust of the paper, in support of the notion that cells that are grouped by similar output have variable physiological parameters. This crucial aspect of the manuscript is not well justified. The criterion for determining cells of similar output is only mentioned in passing with a reference to work of Victor and Purpura without enough detail to evaluate. Since then, many computational studies have been published with highly effective means of identifying cells with highly convergent output. This is underscored by the fact that in Figure 7, the representative traces from these cells are too small to really get a proper feel for this, but the case could easily be made counter to the authors assertion that "spiking outputs are… closely matched within each group". In other words, they don't look that similar to me. If this is based simply on spike number, then it is ignoring the pattern of firing and seems to arbitrarily decide that 2-3 and 4-7 are different groups (why not 2-4 and 5-7?). A much more thorough analysis and justification for "similar outputs" would greatly strengthen the argument of the authors. As this figure is potentially a centerpiece of the study, this is one of the most critical aspects to address.

*Reviewer #3 (Additional data files and statistical comments (optional)):* I do have comments about statistical approaches for this paper, as the manuscript is highly dependent on the analyses for its interpretation.

Figure 2 A rationale for Pearson analysis, which is wedded to a linear relationship, as opposed to Spearman or Kendall Tau, would be appreciated (Pearson does not require normality, because most of the data set is non-normally distributed and rank-based correlation may be more appropriate). The panels in Figure 2 reveal the susceptibility of correlation analysis to outliers and nonlinear relationships. In particular, "Wave decay vs. N spikes (steps)" is a highly suspect correlation, yet is presented as one of the strongest. The power of large sample sizes and the logic of correlation allows for an analysis pathway that can avoid these problems. Specifically, with this level of sample size the authors could remove the tails of the data distribution (perhaps below the 5th and above the 95th percentiles), reanalyze, and see if correlations remain. If a true correlation exists, it will persist among any reasonable subset of the data points.

Figure 3: Nonparametric analyses are performed on these data, but the data are represented with parametric variance. It is unclear what the shaded area represents, or its statistical derivation or value.

Figure 5: If I understand correctly, the authors are making the conclusion that the PCA space changes across developmental time by doing statistics on statistics (an ANOVA of Principal Components). I'm wondering if there are other examples of this approach and whether this derived level of analysis is appropriate. Citations would be appreciated.

Figure 6: It is not clear what the origins of the shaded parts of Figure 6 are? Are these a result of a statistical measure?

[Editors’ note: what now follows is the decision letter after the authors submitted for further consideration.]

Thank you for submitting your work entitled "Multivariate analysis of electrophysiological diversity of *Xenopus* visual neurons during development and plasticity" for consideration by *eLife*. Your article has been reviewed by four peer reviewers, one of whom has agreed to reveal his identity: Nicolas Spitzer. The evaluation has been overseen by a Reviewing Editor and Eve Marder as the Senior Editor.

The reviewers have discussed the reviews with one another and the Reviewing editor has drafted this decision to help you prepare a revised submission.

Summary:

Ciarleglio et al. have followed the changes in an exhaustive list of electrophysiological properties of visual neurons in *Xenopus* during development. This reveals that properties typically diverge over development, but also that visual stimulation can reduce variability.

Essential revisions:

The consensus reached was this is a valuable study well worth publishing, and it was strongly felt that the data set is an important part of the study and should be made public.

There are various options for this. If the data are subject specific then we strongly recommend that the data be deposited in the appropriate location. There is a nice list here: http://www.nature.com/sdata/data-policies/repositories. If the data are heterogeneous and not well suited to a specific repository then one can work with datacite, zenodo, imeji or figshare.

We can of course assist you in this process.

Reviewer #1:

Ciarlegio and colleagues have gone to a lot of trouble to record as many physiological properties as they can from many neurons in a population of *Xenopus* tectum neurons. They compare datasets across developmental periods and a stimulation condition. There appear to be three main conclusions. First, some neural properties change during development and with stimulation, becoming more diverse. Second, that neural properties are highly variable, to the extent that it is hard to predict physiological behavior from 'low level' properties. Third, stimulation appears to decrease variability in the sense that certain features become more correlated.

The first conclusion agrees with a large body of existing data, but does not say anything specific about the possible function of such diversification. For example, do the properties that change make sense from the perspective of how the circuit might function? This kind of question is not really addressed in the manuscript, which instead attempts a more data-driven/assumption free approach. Such an approach needs to have a very transparent analysis and should steer clear of jumping to conclusions. Unfortunately, I found the analysis relied heavily on quite opaque methods that in one case reported clustering (Figure 6) with minute p-values even though no clustering is evident when all the data are plotted together. This reliance on very complicated analysis methods to find patterns in the data made it hard to interpret the findings and assess their veracity.

Another concern which is common to all experimental data that show high variability is the possibility that measurement error and the experimental procedure itself corrupt the readings. The authors chose quite a punishing experimental protocol, and a substantial fraction of cells did not survive. Those that did presumably had stable properties but the only criterion I could find for including a set of recordings was "as long as the nearest IV-curve recording from these cells was stable". This is unacceptably vague – how, for instance, would someone repeat this study using the same criterion? This makes me worry about the quality of the dataset, quite independently of whether it has been over-analyzed. If the dataset (i.e. raw recordings) were to be made available this would help a lot.

Reviewer #1 (Additional data files and statistical comments):

I think the dataset should be made available (raw recordings) and a qualified statistician should look at the manuscript.

Reviewer #2:

The authors have made significant improvements to the manuscript. They have provided better justifications for their conclusions that a population of neurons becomes electrophysiologically more variable during development, that similar spiking behaviors are generated by different constellations of membrane properties and that sensory input reduces electrophysiological diversity.

They have justified their data analysis more fully, in particular checking the main PCA analysis with two standard methods. The case for retaining all the variables makes sense to me. The variance that can be accounted by PCA is now validated through several procedures. The Methods provides a better justification and explanation of the analysis.

They strike a fair balance between clustering neurons into different subtypes and investigating the range of properties of a single type of neurons: the distinction cannot be drawn from the available data. Choosing between the two would be assisted but not resolved by knowing transcriptomes since there is more to differentiation than RNA synthesis, but this is an issue for another day.

Overall the paper is clearer on both experimental and analytical procedures and is more accessible. I am enthusiastic about publication.

*Reviewer #3:*The authors have done a fairly thorough and convincing job of re-packaging these results into a more tractable and consistent narrative, and this has greatly enhanced the paper. There is substantial value in a large and thorough electrophysiological parameter investigation in a vertebrate system, as this is a rare contribution to the literature. In particular what has emerged is an interesting story not so much about degeneracy in mechanisms of output of a single cell type (I believe a flaw in the initial conception of the original manuscript), but rather a thorough and convincing revelation of a continuum of output as revealed by shifting relations among electrophysiological parameters.

A majority of the rebuttal deals with the analysis, and I will admit that I do not have the depth of experience with PCA to contribute much more to this discussion. It seems well conceived and justified to my non-expert eye. The remaining concerns are appropriately addressed.

While the sum total content and conclusion of the paper lacks that clear and concise punchline, in some ways this adds to the charm of the paper. Biology is messy, and in some ways this paper helps to quantify this "messiness" and put it into multiple appropriate and interesting contexts.

I think this study makes a contribution to this area that is worthy of publication.

Reviewer #4:

The authors have taken the reviewers’ comment into account, and have modified the manuscript and added significant analysis accordingly. I have no further comments.

---

## [Author Response]

[Editors’ note: the author responses to the first round of peer review follow.]

In the revised version we have incorporated the reviewers’ suggestions, and done a better job in justifying the various experimental decisions that were made during this study. We have also altered the emphasis of the paper to focus on a different finding, mainly that the diversity of electrophysiological phenotypes increases over development but can be modulated by experience. We have added additional analyses that address the multivariate distribution of cell electrophysiological properties in the original 33-dimentional space, and further support this point, while also allowing us to justify various experimental details. With the altered emphasis and the new data presented we also changed the title of the paper, to better reflect its new message.

For your reference, below is the list of key differences between this new manuscript and the previous version:

• New main point, title, Abstract, and Discussion.

• An in-depth analysis of the dataset validity using the Principal Variables approach was introduced. We now make an informed argument that our dataset presents a productive balance between the independence of individual variables, and the representation of potential interactions between different physiological properties of tectal cells. This new analysis also produced a new panel in Figure 2.

• The main Principal Component Analysis (PCA) is now verified using two different non-linear factorization techniques: Isomapping and Local Linear Embedding. The total share of variance explained by the main PCA is cross-checked and numerically justified in several different ways, including aforementioned non-linear factorization techniques, Bayesian imputation (bootstrapping) of missing variables, and PCA on restricted sets of varialbles pre-sorted by their explanatory power.

• We introduce two new analyses of multivariate distributions of physiological properties in the original 33-dimensional space: quantitative clustering analysis (Agglomerative Clustering coefficients) and within-group PCA as a tool to quantify the strength of multivariate correlations between neural cell properties under different biological conditions. These results are presented in a new figure (Figure 6).

• A clarification of a distinction between true cell types and functional subtypes within the tectal cell population.

• All claims for the "system degeneracy" are removed, while the findings that originally prompted us to make these claims are retained. While we don't make an emphasis on this finding anymore, we nevertheless put a considerable effort into making this part of the paper more convincing, including a better justification of our methods.

• Reported Pearson correlations are verified by Spearman correlations where necessary.

There are also multiple minor changes in variables names, descriptions, and numbering that together contribute to the consistency of the text, and the ease of reading. We also explicitly address several concerns the reviewers had about the choice of variables to include in the dataset, and the quantification approaches we used.

Major Strength:

[…] We are all impressed that you have made a serious effort to collect as many measures as possible from each of a large number of neurons in the frog tectum. In principle, this should constitute an important data set, and could be used to answer a number of interesting questions. And one of the reviewers was, and remains positive about the paper, precisely for this reason. Clearly, the attempt to connect the intrinsic properties and electrophysiological behaviors of the neurons to their underlying component processes is deeply important. But, the devil is in the details, and that is where other reviewers had serious reservations, some of which are expressed in the original reviews and others that surfaced during an extensive set of discussions among the reviewers.

1) The reviewers were unclear about whether your starting assumption was that all of the recorded neurons were of the same cell type: principal neurons of that tectal region?

We now tried to make it clearer in the text that even at the latest of the stages we describe in this paper (stages 48-49 as per (Nieuwkoop and Faber, 1994) almost all neurons in the deep layer of the optic tectum of *Xenopus* tadpoles are considered to belong to one broad cell type, that we refer to as "principal tectal neurons". From studies in adult frogs it is known that eventually, after metamorphosis, optic tectum harbors several types of neurons that differ both in their morphology (Lazar and Szekely, 1967; Lazar, 1973) and function (Grüsser and Grüsser-Cornehls, 1976; Yamamoto et al., 2003; Nakagawa and Hongjian, 2010). In premetamorphic tadpoles however all principal neurons seem to be comparable both in terms of their basic physiological properties (Wu et al., 1996; Khakhalin and Aizenman, 2012; Hamodi and Pratt, 2014) and morphology (Chen et al., 2010; Dong and Aizenman, 2012; Marshak et al., 2012), despite the diversity of their responses to visual stimulation (Podgorski et al., 2012; Khakhalin et al., 2014), and differences in underlying gene expression (Miraucourt et al., 2012). The only obvious non-principal cell type in the developing optic tectum that is known so far are the MT-neurons (Pratt and Aizenman, 2009) that we did not record from.

You made it clear that you were excluding neurons that should have been of a different type, but what evidence is there that all of these neurons should be considered similar? In other words, was it your goal to cluster neurons into reliably different subtypes, or was it your goal to study the range of properties in one type of neuron? The manuscript seems to slide back and forth between these two positions.

In the new version of the manuscript we address this question more directly when discussing clustering of cell properties within experimental groups. Ultimately we cannot tell from our data whether the emerging diversity and clustering of electrophysiological properties at stages 48-49 is due to differential tuning with a single cell type (sometimes referred to as "neuronal subtypes" (McGarry et al., 2010), or a sign of emergent cell differentiation that would only become obvious at post-metamorphic stages. We now better address this ambiguity in the paper.

In the Introduction section:

“Coordinated changes in different physiological properties may contribute to diversification of cell tuning that happens as networks mature, creating and shaping differences in cell phenotypes both between cell types as they emerge (Ewert, 1974; Frost and Sun, 2004; Kang and Li, 2010; Nakagawa and Hongjian, 2010; Liu et al., 2011), and within each cell type in a functional network (Tripathy et al., 2013; Elstrott et al., 2014).”

And in the Discussion:

“Our results indicate that during development, cells in the deep layer of the tectum become more diverse, although at the stages we studied they do not split into distinct non-overlapping cell types that are reported in the tecta of other species and at later stages of development in frogs (Lazar, 1973; Ewert, 1974; Grüsser and Grüsser-Cornehls, 1976; Frost and Sun, 2004; Kang and Li, 2010; Nakagawa and Hongjian, 2010; Liu et al., 2011).”

The best practical way to answer this question would to be to match electrophysiological properties of tectal neurons with their transcriptomes, which is something we hope to do in the future as a follow-up for this study, as we indicate in the last paragraph of the Discussion.

2) Having 33 attributes sounds impressive, but the question is how many of these attributes are independent, and how many are derived properties? To be more specific, we would all agree that the Na and K conductance densities are independent attributes. But, spike frequency and interspike interval are two measures of the same property. Obviously, this latter case is trivial, but there are instances in which two attributes that on first glance may appear to be independent are not. So for example, membrane capacitance and synaptic current might together give you the envelope of the synaptic events? If so, then is it fair to use all three as independent attributes? It is not clear how many truly independent measurements there are among the 33?

This is a very important question, and arguably the main reservation the board of reviewers had about our paper, so in the new version of the paper we added a considerable piece of analysis (Principal Variables analysis) entirely dedicated to this problem. The rationale and significance of this analysis as pertains to your point is as follows:

There are four important considerations we need to keep in mind while working with exploratory analysis (Guyon and Elisseeff, 2003). First, there is no clear cut line between "independent" and "derived" variables in a biological system, especially when complex integrative properties, such as quantification of responses to some type of stimulation are concerned. Most practical variables one could come up with lie somewhere on the spectrum between "fully dependent" (as in a trivial case of frequency and period) and "fully independent".

Second, from the data analysis point of view these "gray zone" semi-independent variables are precisely the variables that are of interest. Obviously, variables that are "too dependent" (in the extreme case – identical) bring no new information in the dataset, and so should be excluded, but the same can be said about fully independent variables. If a cell property is truly statistically independent from all other properties in the set, then except for the very fact of this independency it has nothing new to tell the researcher; it cannot point to an underlying physiological principle (as it does not interact with any other variables), and it cannot help cell classification (for the same reason). From the statistical point of view a truly independent variable is as useless for multivariate analysis as pure noise, which is by definition the epitome of statistically independent.

Third, there is a non-trivial interplay between prior (common sense) and posterior (phenomenological) concepts of variable dependency. For example, some pairs of variables can be assumed to be independent prior to the study (as in your example of Na and K conductances), but shown to be dependent in the dataset (for example in our dataset peak Na and K currents correlate positively, and so are strongly co-dependent). On the contrary, some variables can seem to be very similar based on our limited prior knowledge of the system, but behave surprisingly independently, as it happened for the number of spikes in response to step and cosine injections in our dataset that responded differently to visual stimulation, or two different attempts to quantify overall strength of synaptic inputs to the neuron (through total charge and spontaneous activity) that in practice did not seem to share anything in common.

Finally, the original impetus behind the development of principal component analysis as a method by Galton and Pearson in late 19th – early 20th century was to extract a single underlying parameter (what later became known as first principal component) out of a set of highly correlated variables that were each expected to reflect this underlying parameter without fully representing it. Not only PCA can be extremely useful when applied to datasets of interdependent variables, it relies on the assumption that variables are co-dependent (specifically – linearly correlated). The actual pitfall of applying PCA to diverse multivariate datasets is not the potential dependency of variables per se, but rather the overrepresentation of some aspects of this dependency at the expense of others. For example, when creating our dataset we had to be mindful of including about the same number of variables to quantify each of different aspects of cell physiology (3 for spiking, 3 for spike shape, 2 for spontaneous activity, 2 for synaptic strength, 3 for temporal synaptic properties etc.), lest one of them overwhelmed the rest and skewed the results in its favor.

Based on these considerations, and owing to your clear and constructive feedback, we now include an analysis of overall involvement of different variables in linear correlations with all other variables in the set (the method known as Principal Variable Analysis (Mccabe, 1984) before moving to the factor analysis (Figure 2). We show that our dataset presents a healthy range of variables that each individually explain from 4% to 15% of total variance in the set. None of the variables is too good in predicting the variance; all variables are above the predicted noise level; there is no clear dividing "cliff" in terms of variance explained, but rather a smooth decline across the list; there is a good mix of variables of different biological nature in parts of the plot with higher and lower explanatory power. Overall, this analysis convinced us that our dataset is appropriate for the question, and we hope it helped us to better carry this point across to the reader. This explanation of our rationale for selecting our variables and regarding the principal variable analysis can be found in the text:

“In summary, it can be seen from Figure 2 that no single variable was "too good" in explaining overall variance in the dataset, […] offering a healthy mix of independent and interacting variables (Guyon and Elisseeff, 2003).”

*3) We understand that there is an inherent problem when one is trying to measure many properties from the same neurons, and that it may not be possible to make all measurements perfectly. That said, we don't fully understand why particular choices were made. For example, you do not report the resting potential. And we understand why you made some measurements after bringing all the neurons to -65mV. But that by itself was a decision to not measure the voltage difference between the resting potential and threshold. There are a number of choices of this sort which were not adequately justified in the manuscript, but which might change significantly the take-home messages? You report the number of spikes in response to cosine drive, but from -65mV. These data might look totally different if you had started at the neurons' resting potential (*measurements errors *difficulties of course).*

As tectal neurons are fairly small, the concentration of ions inside the cell matches that in the internal pipette within about 2A3 seconds after the whole-cell patch is established (Khakhalin and Aizenman, 2012). Because of that, a measurement of the "resting membrane potential" in whole-cell mode in our system does not represent the actual resting membrane potential the cell used to have before the patch was established. At the same, we did indirectly quantify Em in the paper by recording the holding current required to bring the cell to −65 mV. From the data analysis point of view, the holding potential is linked to resting membrane potential by a simple formula: Ih = (Em−Ehold)/Rm, and therefore indirectly the value of Em (with the reservations above) was accounted for in our dataset. We now included a corresponding clarification passage in the Methods section of the paper (subheading “Data processing and analysis”); also see more detailed comments on this topic below.

4) A relatively minor consideration that however influences how people approach the paper: the title promises more than the paper delivers and sets it up for criticism.

We now changed both the main message of the paper and its title to better represent our findings. While the fact that the spiking phenotype of each tectal cell is not easily predictable is still presented in the paper, we now think that a more interesting finding is an increase in cell-to-cell tuning complexity in development, and the reshaping of this multivariate tuning after sensory stimulation. We therefore added new analysis of these effects (including quantitative cluster analysis and local PCA analysis), featured these results in a new figure (Figure 6), and put this aspect of our data forward.

*The full individual reviews are below.*Reviewer #1:*1) My main concern is about the poor representativeness of the total variance of the observations by the two principal components used for the PCA (presented in Figure 4, Figure 5, Figure 6, and Figure 7). As the authors mention, the 2 principal components account for 23% of the variance of the observations (15% for PC1 and 8% for PC2). This means that the main part of the variance in the observations (77%) is unaccounted for in this analysis.*

We now included a clarification in the paper, as 23% of variance explained for an incomplete dataset that misses 18% of all possible observations would translate into a larger explained variance for a full dataset, had it been available. We tried to estimate this value by analyzing restricted subsets of data with better representation; applying standard PCA to imputed data, and using alternative approaches to factor analysis, such as Multidimentional Scaling. We conclude that in a dataset without missing values our analysis would have explained about 35 ± 5% of total variance.

The corresponding statement in the Results section reads:

“We concluded that our PCA analysis was adequate for the data, and performed better than local non-linear approaches, with first two principal components explaining 15% and 8% of total variance respectively (this total of 23% of variance explained would have corresponded to about 35% of variance if we had every type of observation in every cell…”

In the Methods section, the following paragraph was added:

“As our dataset was not complete, and 18% of all possible measurements were missing, […] a standard PCA procedure explained on average 35 ± 5% of total variance.”

The choice of analyzing neuronal output by measuring a large number of electrophysiological parameters, and using dimensionality-reduction techniques such as PCA to visualize it is a valid approach if dimensionality reduction preserves most of the information contained in the original measurements.

We would like to respectfully disagree, as PCA and related techniques can be used to find patterns behind interacting values even when the eigenvalues of individual components are relatively small. For example, consider an analysis in which 2 first components of PCA explain 80% of variance in a 10- variable dataset, which would constitute a very robust analysis. Adding 10 more variables of pure noise to this dataset would reduce the amount of total variance explained by first 2 components to approximately 40%, and the components would get more noisy, yet the general pattern of the analysis would be preserved, as the original variables would still have higher loading coefficients than the noisy variables. There are several criteria for the rejection of eigenvalues that are too small, and these criteria are routinely applied to high order principal components as researchers chose the dimensionality of PCA projection. At the very least, they include a requirement of having a value above 1 (that is, explaining more variance than a single normalized column of data would explain), and lying above the "break point" on a scree plot, which we also used in this study (Shabalin and Nobel, 2013).

In our dataset, first two components explained 15% and 8% of variance on incomplete data respectively, which is larger than 5% and 4% expected for two first components on a similarly incomplete dataset that lacks internal structure (consists of uncorrelated normally distributed variables). On a scree plot, the eigenvalue for the 2nd component forms the "break point", and thus can be included in the analysis.

*In the present case, 77% of the variance (most likely including variance in parameters that are essential for defining neuronal output) is lost in the analysis. This problem may arise from several different sources: i) the high number of parameters used as an input for the PCA (33)*

While working on this updated version of the paper, we considered reducing the number of variables included in the PCA analysis, but after considerable internal discussions decided against it. A pre- selection of input variables may make our results seem more convincing, as formally it increases the amount of variance explained. However we feel that in the absence of objective validation, such as a comparison of electrophysiological profile of each cell to its morphological type, pre-sorting and picking of individual variables may not be fully justified. We could conceivably argue in favor of one type of analysis and against the other if one of them proved to be more "productive", and helped to uncover new effects, or hidden structure in the dataset. We found however that pre-selection of variables did not change any of our main findings qualitatively, and thus opted for the inclusion of the original dataset.

The following is a segment about this validation of our PCA approach that we now include in the Methods section of the paper:

“We also attempted restricting the number of variables included in PCA, by pre-screening them based on their Principal Variables rank […] there is no objective post-hoc test to justify the use of one restricted model over another (Guyon and Elisseeff, 2003). We therefore only report it here as another validation of the method.”

ii) the fact that a linear model such as PCA might not be adequate to describe the variance of the parameters analyzed here. Independent of the source of confusion, this means that PCA may not be the adequate model to use in the present study to draw conclusions on the relationships between neuronal output and underlying parameters.

In this revised version of the paper we report two new attempts of using local non-linear approaches to two-dimensional ordination of our data. We concluded that these approaches are less effective than a simple linear PCA. This is the corresponding statement that is now included in the Methods section of the paper:

“While linear approaches to factor analysis, such as PCA or Multidimensional Scaling, are usually considered to be safe go-to methods when noisy weakly correlated data is concerned (Nowak et al., 2003; Sobie, 2009; McGarry et al., 2010), […] these results we concluded that for our data linear factor analysis approach is not only adequate, but also the most appropriate. In all cases testing was performed on centered and normalized data.”

A shorter reference to this validation is also placed in the Results section, prior to the discussion of PCA results:

“We extensively verified the validity of our PCA analysis, comparing it to standard PCA on restricted […] that our PCA analysis was adequate for the data, and performed better than local non-linear approaches.”

In particular, the concept of degeneracy requires that both the target phenotype (here neuronal output) and the underlying parameters are accurately measured (see main point 2) and defined, and that they have relevant relationships. The fact that PCA accounts for only 23% of the variance in "electrical phenotype" is a real concern that hinders the purpose and the conclusions of this study.

Even though our prior conclusions about the degeneracy of spiking phenotype did not rely on the results of our PCA analysis, we now remove all mentions of degeneracy from this paper.

2) My second main concern is about the relevance of some of the electrophysiological parameters measured: some of the parameters are flawed with measurements errors, whilst others lack obvious physiological relevance. The thresholds of the 3 ion currents (Na, KT, and KS) are not measured properly as they are contaminated by reversal potential variations (Na, KT) or are based on a non-sigmoidal fragment of the IV curve (KS). Ion current "thresholds" are usually characterized by precisely defining the half-activation voltages based on conductance vs voltage curves. The measurements of action potential properties in these cells seem to be strongly influenced by variations in the remote location of the spike initiation site (as indicated by the small amplitude of the action potential, around 15-20 mV). In this context, it is difficult to understand whether variations in the kinetics and amplitude of the action potential are mainly attributable to variations in Na and K currents, or variations in the passive properties (i.e. rather related to Cm and Rm) of the compartment located in between the recording site and the spike initiation site.

In our study we were indeed working with the value of membrane potential at which respective active currents became prominent and affected cell spiking dynamics, even though this potential was dependent both on the kinetics of ionic channels, and the distribution of these channels relative to the cell soma and the central process of the cell, where the integration occurs, and from which both the axon initial segment, and the dendritic tree originate. To avoid terminological conflicts, in this new version of the paper we renamed these variables to "I_Na_ activation potential", "I_Ks_ activation potential" and "I_Kt_ activation potential" respectively.

I do not understand the monosynapticity factor.

We changed the description of this factor:

“Traces with longer inter-stimulus intervals (100, 150, 200, 250 and 300 ms) were used to compare the average amplitude […] it was reported as variable #31, "Monosynapticity".”

*3) Because of the first two main concerns, it is difficult to really trust the conclusion about the degeneracy of neuronal output in these neurons. Although there is no doubt that biological systems are "degenerate", demonstrating degeneracy first requires to demonstrate that the variable parameters have a strong influence on the output of the system. As an example, action potential shape is degenerate if i) it relies on variable sodium current density and ii) sodium current density has been demonstrated to have a causal influence on action potential shape. The degeneracy argument is irrelevant when the second condition is not fulfilled: as an example, the fact that synaptic transmission at axon terminals located far away from the soma tolerates large variations in soma size does not tell us anything about the degeneracy of synaptic transmission, until we prove that soma size has a significant influence on synaptic transmission.*

While we feel that the original version of the paper contained at least some justification of the second condition you formulate, as we analyzed which low-level factors affect spiking of tectal cells through a set of competitive linear models, we agree that our case for the true "degeneracy" of physiological properties of tectal cells is rather weak. We therefore remove all mentions of degeneracy from this new version of the paper.

Reviewer #2:*My only suggestion is to change the title to something more positive. Perhaps something like Quantitative analysis of the relation of spiking output to electrophysiological properties reveals substantial degeneracy. I think this will motivate more readers to dip into it.*

After additional analysis of cell clustering in the original 33D space we feel that it would be beneficial to change the main pitch of the paper away from system degeneracy, and towards the changes observed in multivariate variability of cell properties in development, and after sensory stimulation. We therefore have changed the title of the paper to: “Diversity of multivariate tuning of visual neurons in *Xenopus* tadpoles increases in development but is reduced by strong patterned stimulation”

Reviewer #3:

*The authors have provided a substantial data set of electrophysiological properties of tectal neurons of* Xenopus *tadpoles, towards a goal of better understanding the changes in spiking properties of these neurons over development. An additional, and perhaps more striking, point of the paper is the apparent variability of properties that underlie "similar" spiking patterns. In its current form, I'm not sure it has accomplished these goals. There is also a much richer literature in which to root these ideas, particularly in computational arenas, which are largely overlooked.*

We have made every effort to cite the literature relevant to the points raised in this study, but if the reviewer has any specific suggestions we are happy to incorporate them.*Figure 2: The authors seem to have chosen to combine a large dataset of heterogeneous cell types and then perform post-hoc correlation analyses. The rationale for this is unclear, and potentially confounding. The value of the "network" style plot in Figure 2 is minimal. This kind of diagram could potentially be valuable for comparing correlation patterns across groups, but as a standalone, it is difficult to follow any one pair of parameters that the reader would be interested in. The thickness of the lines has no interpretive value without a scale associated. For example, R-values that range from 0.1 to 0.3 would still have a threefold range of thickness, but show minimal correlation regardless.*

The main reason we present this diagram in the paper is to show some of the key correlations in the dataset without creating an extensive list of them, or an additional appendix. With that many variables, only strongest correlations with highest r values are relevant anyway, and we think that the diagram helps to see these large-scale patterns, although they become really obvious later, as we present the PCA results. In our opinion, the circular diagram send a message that the variables form 3 distinct clusters that dominate the dataset in terms of correlations involved, and that compared to these clusters other correlations are weak. This point is emphasized in the Results section of the paper.

Figure 5: Using a PCA, despite the difficulties inherent in not all data being present for all cell types, can be a powerful approach. However, while one can somewhat impute characterizing "spikiness", a meaningful definition of "robustness" is not provided.

We now renamed "Robustness", as a nickname for the second principal component, into "Current density", as it provides a better summary of those variables that are associated with it.

The authors then go on to make the point that the PCA cloud is moving across developmental "space", predominantly in the "spikiness" domain. This is seemingly a bit of an overwrought analysis, when a much more direct measure of "spikiness" (i.e. the spiking of the cells) is provided in Figure 3 and demonstrates in much less convoluted terms that spiking propensity does indeed change in these groups.

While several individual measurements of spikiness changed with development as well, the benefit of PCA as a method is that it can extract the common "signal" from the noisy data, and thus provide a better estimation of the actual phenotypical "Spikiness" of the cell than any single related variable taken alone. That said, we report changes of individual variables in development as well, both in terms of their mean values, and their variability. The PCA analysis further allows us to better understand variability across the cell population, as shown in the new analysis presented in subheading “Factor analysis” Figure 5 and Figure 6.

Figure 7: This is a major thrust of the paper, in support of the notion that cells that are grouped by similar output have variable physiological parameters. This crucial aspect of the manuscript is not well justified. The criterion for determining cells of similar output is only mentioned in passing with a reference to work of Victor and Purpura without enough detail to evaluate. Since then, many computational studies have been published with highly effective means of identifying cells with highly convergent output. This is underscored by the fact that in Figure 7, the representative traces from these cells are too small to really get a proper feel for this, but the case could easily be made counter to the authors assertion that "spiking outputs are… closely matched within each group". In other words, they don't look that similar to me. If this is based simply on spike number, then it is ignoring the pattern of firing and seems to arbitrarily decide that 2-3 and 4-7 are different groups (why not 2-4 and 5-7?). A much more thorough analysis and justification for "similar outputs" would greatly strengthen the argument of the authors. As this figure is potentially a centerpiece of the study, this is one of the most critical aspects to address.

While in this new version of the paper we are now somewhat downplaying the centrality of this figure for the study as a whole, we still think that it is a potentially interesting finding. We used the simple computational approach to spike-train ordination that is described in the work of Victor and Purpura to calculate "distances" between spike-trains of different cells, and represent these spike trains in a 2D plane. We then used same set of distances to pull closest neighbors for several selected reference cells to form clusters of similarly spiking cells. The number of cells per cluster (6) is indeed arbitrary. The clusters we selected are not supposed to represent all possible spiking "phenotypes" we observed, but we used them to test the compactness and noisiness of projections mapping the 2D spiking phenotype space onto 2D spaces of low-level physiological properties of tectal cells.

Here is the updated segment that describes this approach in the paper:

“We then applied a commonly-used standard cost-based metric sensitive to both number of spikes and their timing to quantify similarity between spike-trains of different cells (Victor and Purpura, 1996, 1997), and used multidimensional scaling to represent a matrix of pairwise cell-to-cell distances in a 2D plot (Figure 8). We used this analysis to select groups of cells (6 in each group) that generated similar spike trains in response to current step injections (Figure 8) by pulling 5 nearest neighbors to randomly selected reference cells.”

Reviewer #3 (Additional data files and statistical comments (optional)):*Figure 2 A rationale for Pearson analysis, which is wedded to a linear relationship, as opposed to Spearman or Kendall Tau, would be appreciated (Pearson does not require normality, because most of the data set is non-normally distributed and rank-based correlation may be more appropriate). The panels in Figure 2 reveal the susceptibility of correlation analysis to outliers and nonlinear relationships. In particular, "Wave decay vs. N spikes (steps)" is a highly suspect correlation, yet is presented as one of the strongest. The power of large sample sizes and the logic of correlation allows for an analysis pathway that can avoid these problems. Specifically, with this level of sample size the authors could remove the tails of the data distribution (perhaps below the 5th and above the 95th percentiles), reanalyze, and see if correlations remain. If a true correlation exists, it will persist among any reasonable subset of the data points.*

We agree that in principle Spearman correlation analysis could be more applicable to our dataset, yet we also believe that it is not critical which definition of correlation is used for Figure 2, as we are not interpreting or commenting on any of the weak correlations found in the set, while all strong correlations were significant regardless of whether Spearman of Pearson test was used. As it usually happens when two similar statistical test are compared to each other, the majority of variable pairs were either significantly correlated according to both Pearson and Spearman definitions (n=73 out of 528, after FDR with α=0.05 for both tests) or were not correlated according to both definitions (n=399 of 528). There were however pairs that were correlated in Pearson sense but not in Spearman (n=17), and even more of those that were correlated under Spearman definition but not Pearson (n=39). Because of this fact, we cannot really use the Spearman correlation to "verify" all Pearson correlations. We can use it to verify several top correlations, and we did it, showing that for all Pearson r>0.5 (n=11) Spearman correlations were also significant (see changes in the text below).

Conceivably, we could also migrate all our correlation analysis from Pearson definition to that of Spearman, however we found that it brings only trivial changes to the look of the correlogram at Figure 2, as all main "clusters of correlated variables" remain in place. More importantly, all subsequent analyses in the paper (including principal variable analysis and principal component analysis) rely on the assumption of normality of the data, and utilize standard total variance calculations that are based on Pearson-like products of non-rank-transformed data. As we were aware of this potential issue early on, even before performing any extensive analysis of data we separately ran a validation test by comparing PCA on rank-transformed variables to that on the original data. The validation showed that the results of PCA on rank-transformed variables were not qualitatively different from that of standard PCA, which is reported in the paper (subheading “Statistical Procedures”). Moving to Spearman definition for the sake of correlation analysis would therefore introduce a statistical inconsistency, as it would be equivalent to moving to rank-transformed data for preliminary analysis, yet staying with original data with the final analysis; a choice that may be hard to justify. For similar reasons we did not use tail removal to verify our correlations, as strong correlations remained significant under Spearman definition, which is arguably preferable to data range restriction when outliers are concerned (Linn et al., 1981), while weak correlations were not used for any inference.

Because of these considerations, we decided to keep Pearson correlations in Figure 2. We however removed some of the correlations from Figure 2, as they were not easily interpretable and do not look nice, even though they are still significant after rank transformation. We also include the following statement in the text (Results):

“We verified our Pearson correlation-based analysis calculating Spearman correlation coefficients; after FDR 112 Spearman correlations were significant (as opposed to 90 for Pearson), and all Pearson correlations with r>0.5 (n=11), including all shown in Figure 2, remained significant for Spearman calculation.”

Figure 3: Nonparametric analyses are performed on these data, but the data are represented with parametric variance.

The following statement is now added to the paper:

“As many variables analyzed in this paper were not normally distributed, and to stay consistent, we preferred Mann-Whitney two-sample tests for comparing data between groups; P-values of this test were reported as P_MW_. At the same time, to make referencing and subsequent meta-analysis possible, we always report means and standard deviations in the text.”

That said, if the reviewer still thinks it would be appropriate to report medians and IQR as well, we will be happy to make this change.

It is unclear what the shaded area represents, or its statistical derivation or value.

Corrected description:

“All cell properties that significantly changed with development are shown here as mean values (central line) and standard deviations (whiskers and shading). Transitions between points are shown as shape-preserving piecewise cubic interpolations.”

Figure 5: If I understand correctly, the authors are making the conclusion that the PCA space changes across developmental time by doing statistics on statistics (an ANOVA of Principal Components). I'm wondering if there are other examples of this approach and whether this derived level of analysis is appropriate. Citations would be appreciated.

By definition, the principal components are useful linear combinations of original variables, and as such they can be analyzed by any statistical methods, in the same way as any other formulas or transformations of the original variables can be analyzed. In fact, the very idea of dimensionality reduction approach is to replace the original dataset by its projection in a useful space of lower dimensionality, to perform all subsequent analysis there. This was and remains the main goal of this family of methods, linear and non-linear included, and in most modern applications dimensionality reduction is used not as an exploratory technique, but rather as a preliminary step before main analyses occur. As sample citations, please refer to the introductory section of these texts: (Ghodsi, 2006; Sorzano et al., 2014), page 348 (McKillup, 2011), or page 245 of (Dytham, 2011).

Figure 6: It is not clear what the origins of the shaded parts of Figure 6 are? Are these a result of a statistical measure?

This has been corrected (“Projection of cells from visually stimulated s48A49 animals (black) into PCA space defined by the analysis of naïve dataset, with naïve cells from s48A49 animals shown in blue. Shading shows estimated density kernels for respective groups”).

References:

Chen SX, Tari PK, She K, Haas K (2010) Neurexin-neuroligin cell adhesion complexes contribute to synaptotropic dendritogenesis via growth stabilization mechanisms in vivo. Neuron 67:967-983.

Dong W, Aizenman CD (2012) A competition-based mechanism mediates developmental refinement of tectal neuron receptive fields. J Neurosci 32:16872-16879.

Dytham C (2011) Choosing and using statistics: a biologist's guide, 3rd Edition. Hoboken, NJ: WileyA Blackwell.

Ghodsi A (2006) Dimensionality Reduction A Short Tutorial. Waterloo, Ontario, Canada: Department of Statistics and Actuarial Science. University of Waterloo.

Lazar G, Szekely G (1967) Golgi studies on the optic center of the frog. J Hirnforsch 9:329-344.

Linn RL, Harnisch DL, Dunbar SB (1981) Corrections for range restriction: An empirical investigation of conditions resulting in conservative corrections. Journal of Applied Psychology 66:655.

Marshak S, Meynard MM, De Vries YA, Kidane AH, CohenACory S (2012) Cell-autonomous alterations in dendritic arbor morphology and connectivity induced by overexpression of MeCP2 in *Xenopus* central neurons in vivo. PLoS One 7:e33153.

McKillup S (2011) Statistics explained: an Introductory guide for life scientists, 2nd Edition. Cambridge; New York: Cambridge University Press.

Miraucourt LS, Silva JS, Burgos K, Li J, Abe H, Ruthazer ES, Cline HT (2012) GABA expression and regulation by sensory experience in the developing visual system. PLoS One 7:e29086.

Podgorski K, Dunfield D, Haas K (2012) Functional clustering drives encoding improvement in a developing brain network during awake visual learning. PLoS Biol 10:e1001236.

Sorzano COS, J V, Montano AP (2014) A survey of dimensionality reduction techniques. arXiv preprint arXiv 1403.2877.

Yamamoto K, Nakata M, Nakagawa H (2003) Input and output characteristics of collision avoidance behavior in the frog *Rana catesbeiana*. Brain Behav Evolut 62:201-211.

[Editors' note: the author responses to the re-review follow.]

Essential revisions:The consensus reached was this is a valuable study well worth publishing, and it was strongly felt that the data set is an important part of the study and should be made public.

In our updated submission we have now included all the analysis data as supplementary files. Furthermore, in response to the consensus statement of the reviewers we have uploaded our entire data set to Dryad, including the raw electrophysiological traces of every cell, and the table of extracted parameters and data index file. These data can be accessed at doi:10.5061/dryad.18kk6.

This extensive dataset can be further used to generate physiologically based computational models of the optic tectum, as well as for further analysis of the multiple relationships between various electrophysiological parameters in developing neurons.